# MSTAR: Box-Free Multi-Query Scene Text Retrieval with Attention Recycling

**Liang Yin    Xudong Xie    Zhang Li    Xiang Bai    Yuliang Liu**[*]

Huazhong University of Science and Technology

`{liangyin, xdxie, zhangli, xbai, ylliu}@hust.edu.cn`

## Abstract

Scene text retrieval has made significant progress with the assistance of accurate text localization. However, existing approaches typically require costly bounding box annotations for training. Besides, they mostly adopt a customized retrieval strategy but struggle to unify various types of queries to meet diverse retrieval needs. To address these issues, we introduce Multi-query Scene Text retrieval with Attention Recycling (MSTAR), a box-free approach for scene text retrieval. It incorporates progressive vision embedding to dynamically capture the multi-grained representation of texts and harmonizes free-style text queries with style-aware instructions. Additionally, a multi-instance matching module is integrated to enhance vision-language alignment. Furthermore, we build the Multi-Query Text Retrieval (MQTR) dataset, the first benchmark designed to evaluate the multi-query scene text retrieval capability of models, comprising four query types and $16k$ images. Extensive experiments demonstrate the superiority of our method across seven public datasets and the MQTR dataset. Notably, MSTAR marginally surpasses the previous state-of-the-art model by 6.4% in MAP on Total-Text while eliminating box annotation costs. Moreover, on the MQTR benchmark, MSTAR significantly outperforms the previous models by an average of 8.5%. The code and datasets are available at https://github.com/yingift/MSTAR.

## 1   Introduction

Scene text appears almost everywhere in daily life and is an essential ingredient for image-text searching [6]. Traditional scene text retrieval [9] typically aims to search images based on word or phrase queries and could benefit applications such as handwritten signature retrieval [54, 53] and key frame extraction [35]. However, real-life retrieval needs could be diverse. For instance, people often search for an article with several key words, which cannot be fulfilled with a single word. Additionally, non-ocr visual semantics is also vital for scene text searching. In this work, we study the multi-query scene text retrieval that aims to handle queries of various types within a unified model. This could facilitate applications such as searching disambiguation [31, 16] and visual document indexing [48, 4].

Off-the-shelf scene text retrieval methods [9, 38, 39] achieve text retrieval by explicitly localizing the text and matching the query with scene text instances, as shown in Fig. 1 (a). A straightforward solution is the text spotting methods [22, 20, 49]. They first spot the text in images and match it using edit distance. However, the two divided stages could cause error accumulation between text spotting and query matching. To mitigate this, detection-based methods [38, 41, 39] represent detected ROIs in dense embedding, integrating text detection and similarity learning into an end-to-end framework.

---

[*]Corresponding author

39th Conference on Neural Information Processing Systems (NeurIPS 2025).

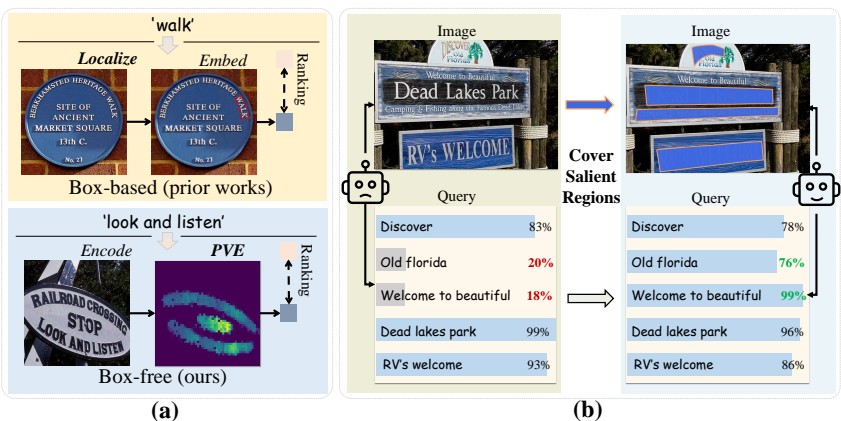

Figure 1: (a) MSTAR achieves scene text retrieval without the aid of box annotations. (b) Image-text matching experiments with VLM [19]. Detailed text instances like "welcome to beautiful" and "old florida" in the image receive lower matching scores. While manually covering salient text regions which receive the higher scores, the model can adaptively recognize the detailed text.

Recently, FDP [51] leverages bounding boxes to guide CLIP [30] in focusing on text regions to achieve accurate retrieval. Despite these advancements, they require expensive box annotations for training. As different retrieval tasks require varying labels, it is very costly to obtain multiple types of bounding boxes, i.e., word-level, text-line, and common object bounding boxes.

Recently, the large-scale box-free pre-training of Vision-Language Models (VLMs) [18, 52] has shown impressive capability on various tasks [1, 55, 50]. To apply box-free methods for scene text retrieval, we conduct image-text matching experiments, as shown in Fig. 1 (b). The observations reveal that box-free models tend to overlook detailed text instances. Moreover, manually masking salient text regions mitigates this issue by enabling the model to adaptively capture image details. More observations are supplemented in the appendix. Inspired by these, we introduce a box-free model termed Multi-query Scene Text retrieval with Attention Recycling (MSTAR). It starts with leveraging pre-trained VLMs for scene text retrieval. To better capture detailed scene text features, the progressive vision embedding is designed to shift attention from salient regions to insalient areas with less attention iteratively. To support diverse retrieval queries, text queries are encoded by the multi-modal encoder with style-aware instructions. A multi-instance matching module is then designed to establish the cross-modal alignment. In this way, MSTAR seamlessly unifies diverse text queries and aligns image-query embeddings without the need of any positional supervision.

To evaluate the performance of multi-query retrieval, we have carefully built a Multi-Query Text Retrieval (MQTR) benchmark. Beyond traditional word and phrase queries, this dataset introduces two additional, valuable retrieval settings: combined query and semantic query. The combined query comprises several discontinuous key texts (words or phrases) to enable more precise retrieval. Semantic queries, on the other hand, are image descriptions that require understanding both scene text and its non-ocr visual context [45]. In total, the MQTR dataset includes four styles of queries (word, phrase, combined, and semantic) and 16000 images.

Experiments reveal that existing models struggle to simultaneously handle four types of query on the challenging MQTR dataset. In contrast, our proposed MSTAR performs well in multi-query retrieval. Notably, MSTAR impressively surpasses previous work by an average of 8.5% in MAP. Additionally, on seven public retrieval datasets, MSTAR demonstrates competitive performance compared to state-of-the-art box-based methods while significantly reducing annotation costs. Specifically, MSTAR outperforms FDP [51] by 6.4% in MAP on the widely used Total-Text dataset.

We summarize the main contributions as follows: (1) We propose MSTAR, the first box-free method designed for scene text retrieval, which eliminates box annotations and achieves multi-query retrieval. (2) We collect MQTR, a comprehensive benchmark with four types of queries and 16,000 images for evaluating multi-query scene text retrieval. (3) Experiments on seven public datasets and the MQTR benchmark demonstrate the advantages of MSTAR.

## 2 Related Work

**Scene Text Retrieval Datasets.** Existing scene text retrieval datasets primarily focus on single-type retrieval such as word and phrase. The IIIT Scene Text Retrieval dataset [27] is a large-scale dataset dedicated to word retrieval comprising 10k images. The COCOText Retrieval dataset is derived from 7k natural images in the COCOText dataset [37]. In addition, several smaller but well-annotated datasets [40, 7] are commonly utilized for evaluating word retrieval performance. The Chinese Street View Text Retrieval dataset [38] contains 23 Chinese queries and 1,667 scene text images from Chinese street views. Besides these word retrieval datasets, the Phrase-level Scene Text Retrieval [51] dataset includes 36 frequently used phrase queries and 1,080 images. The CSVTRv2 [39] dataset supports partial and text-line queries. However, these datasets do not comprehensively support the evaluation of tasks such as combined retrieval and semantic retrieval. In contrast, our MQTR dataset can support four types of query to satisfy diverse needs in real-world applications.

**Scene Text Retrieval Methods.** Existing methods generally adopt a paradigm that first localize text regions and then match with the query. In the early attempts, Mishra *et al.*[27] proposed identifying approximate character locations and indexing words using spatial constraints. More recent approaches try to integrate the two process into an end-to-end trainable framework. Gomez *et al.*[9] proposed to combine the detected proposals with the Pyramidal Histogram of Characters [2]. Wang *et al.*[38] unified the text detector with a cross-modal similarity model into an end-to-end framework. Its updated version [39] proposed the RankMIL and DPMA algorithms to address the partial scene text retrieval problem. Wen *et al.*[41] transformed cross-modal similarity into uni-modal similarity using image templates. Zeng *et al.*[51] utilized CLIP [30] for scene text retrieval and incorporated box supervision to localize text regions. However, these methods require expensive bounding box annotations for training.

In addition to the specifically designed retrieval methods, text spotters can also be applied to retrieval tasks. They first spot text instances from images and then rank with edit distance. Traditional text spotters with boundary supervision [22, 20, 44] can achieve accurate results. There are also point-supervised [29, 23] and transcription-only supervised methods [36, 42, 43], which yield limited performance. Above all, the separation of text spotting and query matching often leads to error accumulation, and text spotters struggle to handle multi-query retrieval.

Unlike above methods, our MSTAR achieves text retrieval without the bounding-box supervision and harmonize various data labels for multi-query scene text retrieval.

## 3 Method

### 3.1 Overview

The overall architecture of MSTAR is depicted in Fig. 2. Given a scene text image, the vision encoder extracts image features $f_0$, and the Progressive Vision Embedding module progressively captures the scene text features as vision embeddings $E_V$. Simultaneously, text queries are encoded as text embeddings $E_T$ by the multi-modal encoder from BLIP-2 [19] with style-aware instructions. The $E_V$ and $E_T$ are then fed into the Multi-Instance Matching module to align the cross-modal embeddings optimized with contrastive loss. Additionally, a re-ranking process is incorporated by inputting both image features and text queries into the multi-modal encoder to compute one-to-one matching scores. During training, MSTAR is optimized using both contrastive loss and image-text matching loss. For inference, image-text pairs are initially ranked by the cosine similarity, and the top K images are further matched with re-ranking process.

### 3.2 Progressive Vision Embedding

Vision-Language Models (VLMs) pre-trained on image-caption pairs often focus more on salient visual concepts such as a red circle [33, 47]. However, the detailed and subtle scene text instances typically appear in various insalient regions of natural scenes. As depicted in Fig. 1 (b), the "old florida" and "welcome to beautiful" are overlooked as the model focus on more salient regions. This problem leads to a high miss rate for small text scenarios in scene text retrieval tasks. To mitigate this, we propose a Progressive Vision Embedding (PVE) approach to extract visual embeddings.

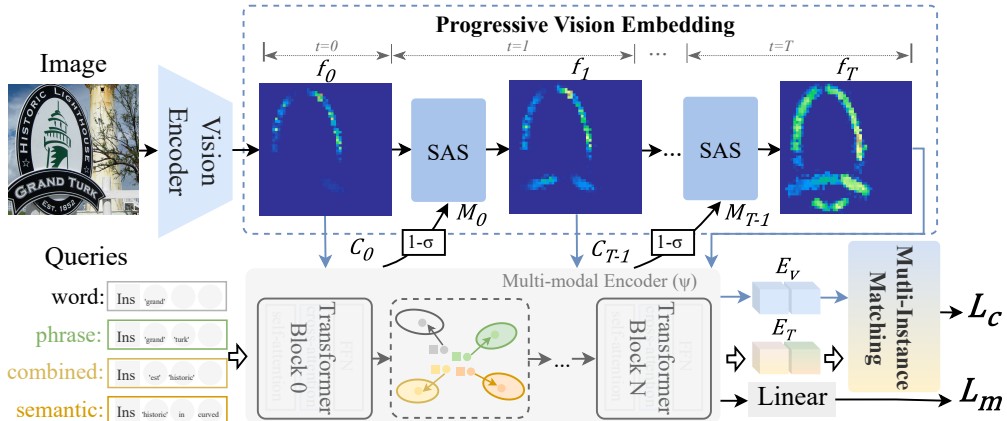

Figure 2: Overview of MSTAR. MSTAR is built upon four key components: a vision encoder $\phi$, the Progressive Vision Embedding (PVE), the multi-modal encoder $\psi$, and the multi-instance matching module (MIM). PVE incorporates image features $f_t$ and the mask $M_t$ derived from cross-attention map $C_t$, progressively shifting attention from salient features to fine-grained regions.

Given a scene text image $I \in \mathbb{R}^{H \times W \times 3}$, the vision encoder $\phi$ encodes $I$ into initial image features, denoted as $f_0 \in \mathbb{R}^{L \times D}$. Then a two-layer MLP is used to align the hidden dimensions of $\phi$ and the multi-modal encoder $\psi$ stacked with transformer blocks. Subsequently, $\psi$ is leveraged to capture the scene text vision embeddings from $f_0$ with learnable queries $Q_l \in \mathbb{R}^{Q \times d}$. The initial vision embeddings are denoted as $E_V^0 \in \mathbb{R}^{Q \times d}$.

Since the model tends to focus on salient image features (visualized in Fig. 2), $E_V^0$ struggles to fully capture the detailed text instance representation. To force the model to focus on less salient features, we propose the Salient Attention Shift (SAS) module. Motivated by observations in Sec. 1, the SAS uses a mask-attention layer to automatically shift image attention. Unlike previous methods [5] that use ground truth as supervision, the mask in our approach is derived from the cross-attention layers of the multi-modal encoder $\psi$. Concretely, we first calculate the mean of the cross-attention weights of $\psi$ as the cross-attention map $C_{t-1}$. Then $C_{t-1}$ is binarized with a binarization algorithm $\sigma$ and inverted to obtain a binary mask $M_{t-1}$. This is formulated in Eq. 1.

$$M_{t-1} = 1 - \sigma(C_{t-1}), \tag{1}$$

where the $\sigma$ consists of a thresholding with low threshold for coarse filtering, a marker-based watershed algorithm for precise binarization, and a connected components algorithm to avoid over-segmentation. In the t-th step, the SAS refines image features as follows:

$$f_t = \mathcal{S}(f_{t-1}, \text{mask} = M_{t-1}), \tag{2}$$

where $\mathcal{S}$ denotes multi-head self-attention, $f_{t-1}$ is image features and $M_{t-1}$ is the binary mask. For each pixel $M_{t-1}^{i,j}$, a value of 0 indicates that the corresponding image features of the previous iteration received high attention, while a value of 1 indicates lower attention. With $M_{t-1}$, the SAS learns to adaptively reduce the weight of salient features and focus more on the neglected features.

In each iteration, the SAS dynamically renews the image features, as illustrated in Fig. 2. Then the multi-modal encoder $\psi$ embeds the $f_t$ to $E_V^t$. Then the vision embeddings $\{E_V^0, E_V^1, \dots E_V^T\}$ are concatenated to $E_V \in \mathbb{R}^{(T+1)Q \times d}$, where $T$ is the maximum recurrent steps.

### 3.3 Instruction Aware Text Representation

In unified training of multi-query scene text retrieval, the difference of query styles (e.g., format and characteristics) could cause semantic discrepancy. For example, the phrase query contains several continuous words with linguistic semantics, but the combined query contains several discrete key words. To harmonize the representation of these text queries, a short style-aware text instruction is introduced to guide the embedding for each type of query. The process is formulated as follows:

$$E_T = \psi(\text{Concate}[T_i, T_Q]), \tag{3}$$

where $\psi$ denotes the multi-modal encoder, $T_i$ represents instructions and $T_Q$ denotes text queries. The instructions prompt the $\psi$ to distinguish query types. As shown in Fig. 2, queries of each style are encoded into different representation space during training. To speed up training, all of the text queries (words, phrases, combined, and semantic queries) paired to the image are encoded altogether. The $\psi$ encodes the queries into text embeddings $E_T \in \mathbb{R}^{N \times d}$ which consist of single-word embeddings $E_w \in \mathbb{R}^{N_w \times d}$ and multi-word embeddings $E_m \in \mathbb{R}^{N_m \times d}$. The N is the total number of text queries paired to the image. The $N_w$ and $N_m$ denote the number of single-word queries and multi-word queries, respectively.

### 3.4 Multi-Instance Matching

After obtaining the vision embeddings $E_V \in \mathbb{R}^{(T+1)Q \times d}$ and text embeddings $E_T \in \mathbb{R}^{N \times d}$, the problem is to build the one-to-one alignment for the multi-type and multi-instance embeddings. The previous study either aggregate the multiple vision embeddings into one embedding [19] or adopts the late interaction [15, 8]. However, due to the implicit matching mechanism, these strategies need massive training for vision-language alignment.

To mitigate these, we propose the Multi-Instance Matching (MIM) module to explicitly assign the one-to-one matching relations for vision-language embeddings. MIM comprises two parallel branches for processing single-word and multi-word queries, respectively. In the single-word branch, the Hungarian matching algorithm [17] is exploited to explicitly assign the one-to-one matching relation between $E_w \in \mathbb{R}^{N_w \times d}$ and $E_V \in \mathbb{R}^{(T+1)Q \times d}$. Specifically, we first construct a cosine similarity matrix of size $N_w \times (T+1)Q$. Since $N_w$ is typically unequal to $(T+1)Q$, we pad the matrix with zeros to create a square matrix. Finally, the first $N_w$ rows of the results are used for one-to-one correspondences.

In the multi-word branch, since the multi-word queries contain abundant semantic information, a light-weight cross-attention layer is used to aggregate the vision features under text constraint in the second branch. This process is formulated as Eq. 4.

$$E_{vt} = \mathcal{F}((\mathcal{C}(\mathcal{Q} = E_w, \mathcal{K}, \mathcal{V} = E_V))), \tag{4}$$

where $\mathcal{F}$ denotes a feed-forward network and $\mathcal{C}$ denotes multi-head cross-attention. The two branches adaptively shift to cope with different types of queries, i.e., word retrieval relies solely on the first branch, while multi-word queries with the second. Thanks to this flexible alignment approach, mixed training data labels can be effectively leveraged for training multi-query retrieval models.

### 3.5 Optimization

MSTAR is optimized with both contrastive learning and image-text matching task. Contrastive learning enables the model to separately encode vision embeddings and text embeddings. A dual contrastive loss $\mathcal{L}_c$ aligns the vision and text embeddings.

$$\mathcal{L}_c = \alpha \mathcal{L}_{t2v} + \mathcal{L}_{v2t}, \tag{5}$$

where $\alpha$ is a hyperparameter to maintain the numerical approximation equivalence of the two losses. Since the number of queries is usually greater than the number of images, $\alpha$ is set to 1.5 in our implementation. The image-text matching process simultaneously feeds image features and text queries into the multi-modal encoder $\psi$. An image-text matching score is then computed using a linear layer with a two-cell output. This score is optimized using a cross-entropy matching loss $\mathcal{L}_m$. The overall loss is the sum of $\mathcal{L}_c$ and $\mathcal{L}_m$.

## 4 Experiments

### 4.1 Multi-Query Text Retrieval Dataset

To comprehensively evaluate the performance of models on multi-query scene text retrieval, we carefully build the Multi-Query Text Retrieval (MQTR) dataset. The MQTR dataset includes four sub-tasks: word, phrase, combined, and semantic retrieval. The construction of this dataset leverages well-annotated public datasets [37, 21, 13, 14, 7, 34, 24, 51], along with images obtained from Google Image Search. The word, phrase, and combined subsets each contain 5,000 images and the 200 most frequently occurring queries. The semantic subset consists of 1,000 images and 25 queries collected from the web. The semantic subset was manually collected with 10-15 positive images and an equal

| Dataset | Venue | Word | Phrase | Combined | Semantic | Q. Num | Images |
|---------|-------|------|--------|----------|----------|--------|--------|
| Total-Text [7] | IJDAR'20 | ✓ | ✗ | ✗ | ✗ | 60 | 300 |
| CTW [21] | PR'19 | ✓ | ✗ | ✗ | ✗ | 100 | 500 |
| IC15 [14] | ICDAR'15 | ✓ | ✗ | ✗ | ✗ | 100 | 500 |
| CTR [37] | Arxiv'16 | ✓ | ✗ | ✗ | ✗ | 500 | 7196 |
| STR [9] | ECCV'18 | ✓ | ✗ | ✗ | ✗ | 50 | 10k |
| CSVTR [38] | CVPR'21 | ✗ | ✓ | ✗ | ✗ | 23 | 1667 |
| PSTR [51] | ACM MM'24 | ✗ | ✓ | ✗ | ✗ | 36 | 1080 |
| MQTR | - | ✓ | ✓ | ✓ | ✓ | 625 | 16k |

Table 1: Statistics of public scene text retrieval datasets and our MQTR dataset in terms of query types, number of queries, and number of images.

| Method | Venue | AVG. | Word | Phrase | Combined | Semantic |
|--------|-------|------|------|--------|----------|----------|
| *Box Based* | | | | | | |
| ABCNet [22] | TPAMI'21 | 24.13 | 26.14 | 15.15 | 36.47 | 18.74 |
| MaskTextSpotter [20] | ECCV'20 | 32.43 | 46.72 | 27.53 | 29.08 | 26.37 |
| TDSL [38] | CVPR'21 | 58.25 | 69.11 | 40.83 | 72.71 | 50.36 |
| Deepsolo [49] | CVPR'23 | 52.04 | 67.54 | 25.68 | 72.14 | 42.79 |
| TG-Bridge [11] | CVPR'24 | 54.09 | 69.89 | 30.21 | **75.53** | 40.73 |
| *Box Free* | | | | | | |
| SPTSv2 [23] | TPAMI'23 | 35.18 | 33.56 | 21.24 | 50.76 | 35.16 |
| BLIP-2 [19] | PMLR'23 | 36.13 | 17.31 | 32.76 | 25.80 | 68.63 |
| SigLIP [52] | CVPR'23 | 36.06 | 17.81 | 32.88 | 21.81 | 72.23 |
| BLIP-2 (FT) [19] | PMLR'23 | 58.11 | 58.09 | 42.23 | 60.84 | 71.24 |
| MSTAR | - | **66.78** | **73.27** | **44.22** | 74.48 | **75.14** |

Table 2: Evaluations of MAP% on MQTR. FT denotes fine-tune. The best results are shown in **bold**.

number of hard negative samples for each query. The hard negatives samples refer to images that contain three types of objects: (1) visual elements with semantics similar to the query (e.g., an apple and the word "apple"), (2) text instances with similar shapes, and (3) text instances with similar meanings. The inclusion of hard negatives poses additional challenges by introducing visually and textually confounding samples, thus assessing the capacity of retrieval models to distinguish visual semantics from OCR-based semantics. As demonstrated in Tab. 1, our MQTR dataset is the first comprehensive benchmark to support four types of query in scene text retrieval. You can refer to the appendix for more construction details and images samples from the datasets.

## 4.2 Implementation details

The visual encoder $\phi$ is initialized from ViT-Base-512 of SigLIP [52]. The multi-modal encoder $\psi$ is initialized from BLIP-2 [19]. The number of query tokens $Q_1$ is set to 64 with interpolation, which is consistent with the setting of the vanilla BLIP-2 in our comparison experiments. The weights of the MLP, SAS, and MIM modules are randomly initialized. The MSTAR model was trained on four NVIDIA A800 GPUs and evaluated on a single GPU, using the AdamW optimizer [25]. A multi-stage training is adopted with progressive resolution increasing from 512×512, 640×640, to 800×800. For re-ranking, the top 2% of images are selected from the initial retrieval results. For instance, in a dataset containing $10k$ images, only the top 200 images are used for re-ranking.

## 4.3 Multi-Query Scene Text Retrieval

To enable multi-query retrieval, we collect a training dataset consisting of $95k$ images. First, $50k$ synthetic images with word transcriptions are leveraged from SynthText-900k [9]. Then $20k$ real images containing captions are collected from TextCap [34]. We use labels with both image captions and text transcriptions acquired with Rosetta [3]. In addition, we have

| BLIP-2 [19] | TDSL [38] | SigLIP [52] | FDP [51] | MSTAR |
|-------------|-----------|-------------|----------|-------|
| 85.49 | 89.40 | 89.56 | 92.28 | **95.71** |

Table 3: Comparisons on Phrase-level Scene Text Retrieval dataset [51].

| Method | Venue | SVT | STR | CTR | Total-Text | CTW | IC15 | Avg. | FPS |
|---|---|---|---|---|---|---|---|---|---|
| | | *Box Based* | | | | | | | |
| Mishra *et al.* [27] | ICCV'13 | 42.70 | 56.24 | - | - | - | - | - | 0.1 |
| Jaderberg *et al.* [12] | IJCV'16 | 86.30 | 66.50 | - | - | - | - | - | 0.3 |
| Gomez *et al.* [9] | ECCV'18 | 83.74 | 69.83 | 41.05 | - | - | - | - | 43.5 |
| Mafla *et al.* [26] | PR'21 | 85.74 | 71.67 | - | - | - | - | - | 42.2 |
| TDSL [38] | CVPR'21 | 89.38 | 77.09 | 66.45 | 74.75 | 59.34 | 77.67 | 74.16 | 12.0 |
| Wang *et al.* [39] | TPAMI'24 | - | 81.02 | **72.95** | - | - | - | - | 9.3 |
| Wen *et al.* [41] | WSDM'23 | 90.95 | 77.40 | - | 80.09 | - | - | - | 11.0 |
| FDP-RN50×16 [51] | ACM MM'24 | 89.63 | **89.46** | - | 79.18 | - | - | - | 11.8 |
| | | *Box Free* | | | | | | | |
| BLIP-2 (FT) [19] | PMLR'23 | 88.73 | 85.40 | 45.75 | 77.20 | 82.33 | 55.13 | 72.42 | 37.2 |
| MSTAR | - | **91.31** | 86.25 | 60.13 | 85.55 | 90.87 | 81.21 | 82.56 | 14.2 |
| MSTAR (+re-rank) | - | 91.11 | 86.14 | 65.25 | **86.96** | **92.95** | **82.69** | **84.18** | 6.9 |

Table 4: Comparisons with scene text retrieval methods of MAP% on 6 public word retrieval datasets. The best results are shown in **bold**, and the second results are underlined.

| Method | Venue | SVT | STR | CTR | Total-Text | CTW | IC15 | Avg. | FPS |
|---|---|---|---|---|---|---|---|---|---|
| | | *Box Based* | | | | | | | |
| ABCNet [22] | TPAMI'21 | 82.43 | 67.25 | 41.25 | 73.23 | 74.82 | 69.28 | 68.04 | 17.5 |
| MaskTextspotterV3 [20] | ECCV'20 | 83.14 | 74.48 | 55.54 | 83.29 | 80.03 | 77.00 | 75.58 | 2.4 |
| Deepsolo [49] | CVPR'23 | 87.15 | 76.58 | 67.22 | 83.19[*] | 87.67[*] | 82.80[*] | 80.77 | 10.0 |
| TG-Bridge [11] | CVPR'24 | 87.23 | 81.30 | **70.08** | **87.11**[*] | 88.39[*] | **83.55** [*] | 82.94 | 6.7 |
| | | *Box Free* | | | | | | | |
| SPTSv2 [23] | TPAMI'23 | 78.08 | 62.11 | 48.39 | 73.61[*] | 83.30 [*] | 66.27[*] | 68.63 | 7.6 |
| MSTAR | - | **91.31** | **86.25** | 60.13 | 85.55 | 90.87 | 81.21 | 82.56 | 14.2 |
| MSTAR (+re-rank) | - | 91.11 | 86.14 | 65.25 | 86.96 | **92.95** | 82.69 | **84.18** | 6.9 |

Table 5: Comparisons with mainstream scene text spotting methods. **\*** indicates that the model has been fine-tuned on the corresponding training set. The best results are highlighted in **bold**, and the second results are underlined.

synthesized $25k$ images with phrase transcrip-
tions using the synthesis engine [10]. Additionally, word or phrase annotations are utilized to form nonrepeated combined queries for images containing over one text instance. More training details can be found in the appendix.

On the MQTR dataset, we perform evaluation with both the representative box-based models and box-free models for multi-query scene text retrieval. For text spotting methods, we use the normalized edit distance to measure query-image matching scores following [38]. To evaluate the models that can only handle word queries, we calculate the mean similarity of each word as the image-text scores for multi-word queries. The codes and weights are acquired from their official repositories.

**Evaluation on multi-query scene text retrieval.** As the results reported in Tab. 2, box-based methods [38, 49, 11] typically perform better on word queries and combined queries, which demand fine-grained scene text perception. However, these box-based models cannot leverage the rich linguistic semantics for phrase and semantic retrieval. Compared to box-based methods, MSTAR outperforms TG-Bridge[11] by 3.38% and Deepsolo[49] by 5.73% in word retrieval. On the other hand, VLMs such as BLIP-2 (ViT-L) [19] and SigLIP (ViT-B-512) [52] perform better on phrase and semantic queries but struggle in word and combined retrieval. Compared to them, MSTAR outperforms SigLIP by 11.34% on phrase retrieval and 2.91% on semantic retrieval. In terms ofms of overall results, our MSTAR obtains an improvement of 8.53% over previous works on average. These comparison results show the great advantages of our MSTAR on multi-query retrieval.

Additionally, MSTAR is also evaluated on the phrase-level scene text retrieval dataset [51]. As shown in the Tab. 3, our MSTAR achieves 95.71% of MAP on the benchmark.

| Ins | MIM | PVE | CTR | SVT | STR | Total-Text | CTW | IC15 | MQTR |
|---|---|---|---|---|---|---|---|---|---|
| ✗ | ✗ | ✗ | 52.87 | 90.07 | 81.57 | 82.32 | 87.28 | 76.71 | 65.79 |
| ✓ | ✗ | ✗ | 54.65 | 90.70 | 82.81 | 83.19 | 88.96 | 77.15 | 66.15 |
| ✓ | ✓ | ✗ | 55.77 | 91.02 | 85.00 | 84.01 | 90.31 | 79.23 | 65.69 |
| ✓ | ✓ | ✓ | 60.13 | 91.31 | 86.25 | 85.55 | 90.87 | 81.21 | 66.78 |

Table 6: Ablation studies on instruction, multi-instance matching, progressive vision embedding, denoted as Ins, MIM, PVE, respectively.

## 4.4 Word-level Scene Text Retrieval

In this part, we present comparison experiments on word-level retrieval. The model is trained on $100k$ images randomly sampled from SynthText-900k [9] and 5k images from MLT-5K [28] dataset. The evaluation setting keeps the same as the previous method [38]. Note that Deepsolo [49], TG-Bridge [11], and SPTSv2 [23] are well fine-tuned on the training set of Total-Text, CTW and IC15 dataset.

**Comparisons with text retrieval methods.** We conduct comprehensive evaluations on the test sets of six public datasets, the results are presented in Tab. 4. Compared to FDP-RN50×16 [51], our MSTAR achieves an improvement of 1.68% on SVT and 6.37% on Total-Text. While MSTAR demonstrates a slightly lower performance on the STR dataset, it eliminates the cost of expensive bounding-box for training. Compared to TDSL [38], our method significantly outperforms the method across five datasets. Notably, our MSTAR surpasses TDSL by 9.16% on STR, 10.80% on TotalText, and 31.53% on CTW. These results verify the robust capabilities of our model. However, on the CTR dataset that contains extreme small text, our method underperforms TDSL. This is due to the absence of precise box supervision of our method, which is a common problem for box-free methods. Overall, MSTAR outperforms TDSL by an average 8.40% in MAP across six datasets. To further enhance performance, we re-rank the top 2% of retrieved images by jointly feeding the text queries and images into the model. This re-ranking strategy improves performance by an additional 1.56% in MAP.

**Comparisons with text spotting methods.** Tab. 5 shows the comparison results with scene text spotting methods. Compared to box-free method, our model significantly outperforms SPTSv2 with an average improvement of 13.93% in MAP. To further verify the advantages of MSTAR, we compare it with state-of-the-art box-based text spotting methods. The results in Tab. 5 indicate that MSTAR achieves competitive performance with the advanced TG-Bridge on average. Moreover, MSTAR offers over twice the inference speed compared to TG-Bridge (14.2 FPS vs. 6.7 FPS) due to the absence of the text detection module. These results show that our model achieves competitive performance with leading fully supervised models while eliminating the bounding box for training.

## 4.5 Ablation Study

To validate the effectiveness of each component, comprehensive ablation studies were conducted.

**Ablation study on three core components.** The overall results are reported in Tab. 6. We begin by directly training the vision encoder, MLP and multi-modal encoder with standard contrastive learning. The results suggest that this baseline performs poorly on the fine-grained recognition capability, especially on the CTR and IC15 datasets which features small text. Then instructions are adopted to prompt the model to encode queries which leads to improvement. Subsequently, the MIM is added, which leads to a significant improvement of 2.19% on STR and 1.35% on CTW. However, a slight performance drop of 0.46% is observed on the MQTR dataset. This may be because of the use of Hungarian matching, which is effective for word-level queries and instance-level alignment, introduces confusion when handling more complex queries.

Lastly, we incorporate PVE which is designed to improve the retrieval performance of the insalient objects such as small and detailed text instances. The results show a substantial improvement on CTR (4.36%) and IC15 (1.98%), validating the effectiveness of PVE in small text scenarios.

**Ablation study on the binary $\sigma$ algorithm.** We investigate three variants of $\sigma$, as the results reported in Tab. 7. 1) Zero Padding: A binary mask with all zero values is used, which means the mask does not constrain the image attention. The results show only a slight improvement on CTR (0.91%), which is probably due to the effectiveness of the progressive representation strategy. 2) The second

| $\sigma$ | CTR | Total-Text | IC15 |
|---|---|---|---|
| No PVE | 55.76 | 84.01 | 79.23 |
| Zero Pad | 56.67 | 83.79 | 79.27 |
| TH+CC | 59.66 | 85.17 | 80.17 |
| TH+WS+CC | 60.13 | 85.55 | 81.21 |

Table 7: Ablation studies on $\sigma$ algorithm. TH denotes ThresHolding, CC denotes Connected Components, and WS denotes WaterShed algorithm.

choice of $\sigma$ is composed of thresholding and connected components. This variant can produce a mask to guide the SAS module to refine attention focus. However, it requires repeated tuning of the thresholds. 3) The third choice is to first apply coarse filtering to the background with a low threshold and then obtain precise binarization results using the watershed algorithm. This approach eliminates the need for complex hyperparameter tuning and achieves substantial improvements, including 4.37% on CTR, 1.54% on Total-Text dataset, and 1.98% on the IC15 dataset. We adopt the third variant for our model.

**Ablation study on the number of iteration steps T in PVE.** We investigate the impact of the number of recurrent steps T in PVE. As the results presented in Tab. 8, performance improves significantly as T increases from 0 to 1. As T increases from 1 to 3, we can also observe noticeable improvement on the three datasets. Since the inference speed decreases with the recurrent steps increases, we adopt T=1 for the final model to balance efficiency and effectiveness.

| T | CTR | Total-Text | CTW | FPS |
|---|---|---|---|---|
| 0 | 55.76 | 84.01 | 90.31 | 16.5 |
| 1 | 60.13 | 85.55 | 90.87 | 14.2 |
| 2 | 60.47 | 86.68 | 90.95 | 12.9 |
| 3 | 60.87 | 87.66 | 91.24 | 11.2 |

Table 8: The impact of the recurrent steps T.

## 5   Discussion

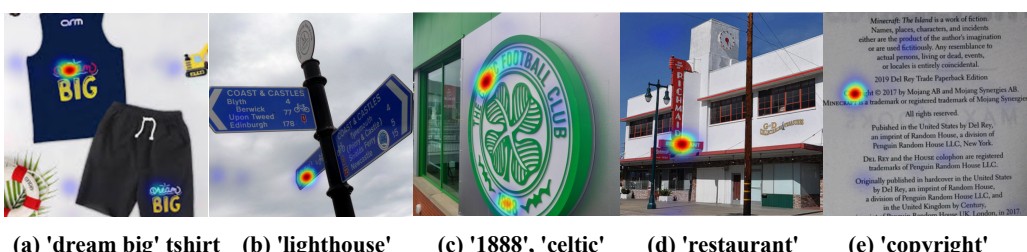

(a) 'dream big' tshirt     (b) 'lighthouse'     (c) '1888', 'celtic'     (d) 'restaurant'     (e) 'copyright'

Figure 3: Visualization of the text localization of our MSTAR. The image shows the localization of (a) semantic, (b) phrase, and (c) combined query, as well as (d) curved and (e) dense word instances.

**Application of MSTAR for text localization.** To further validate the effectiveness of MSTAR, we use it for text localization using Grad-CAM [32]. As illustrated in Fig. 3, MSTAR can localize different types of queries. For example, MSTAR accurately identifies the target text instance "dream big" on a t-shirt, distinguishing it from the "dream big" of the shorts in Fig. 3 (a). In addition, MSTAR can also accurately localize curved and dense text instances. As Fig. 3 (e) shows, MSTAR successfully identifies the word "copyright" within a document page image. These results demonstrate that MSTAR can accurately localize text instances without box supervision.

**Discussion about the pre-trained model.** We discuss how the pre-trained model affects performance on the scale of parameters and data. We tested different variants of CLIP [46], BLIP [18], BLIP-2 [19], and SigLIP [52] on the CTR dataset. The results show that despite the massive pre-training, the models struggle to achieve retrieval on small text instances. Details are provided in the appendix.

**Limitations.** Although our method shows evident improvements in fine-grained scene text retrieval, it still lags behind box-based methods when handling extremely small and dense text instances. This

limitation stems from insufficient positional supervision, which is a common limitation to box-free approaches.

## 6 Conclusion

This work investigates the fundamental OCR task of scene text retrieval via a novel box-free paradigm. It incorporates PVE to shift visual attention to fine-grained scene text. Our model demonstrates competitive retrieval performance with state-of-the-art box-based methods while significantly reducing annotation cost. Moreover, for the first time, we study the multi-query scene text retrieval enabling broader real-world applications. Experiments show that neither box-based methods nor general cross-modal retrievers can handle such a challenging task effectively, while our method can serve as a strong baseline for this task. This study is expected to inspire future research on weakly supervised pre-training for visual document foundation models.

## Acknowledgements

This work was supported by the National Key Research and Development Program of China (Grant 2022YFC3301703) and the NSFC (Grants 62206104, 62225603).

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

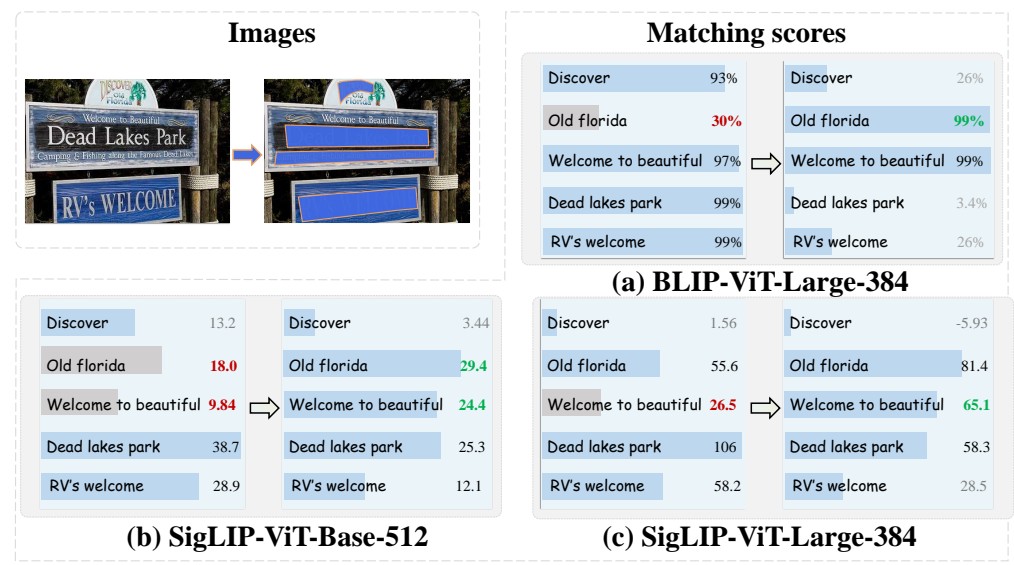

Figure 4: Qualitive analysis of VLMs to process in-salient text instances , which is introduced in Sec. 1, (a) BLIP-ViT-Large-384, (b) SigLIP-ViT-Base-512, and (c) SigLIP-ViT-Large-384.

## A    Further Analysis of Vision Language Models

In this section, we first provide further analysis of Vision Language Models (VLM) in processing in-salient text instances introduced in Sec. A.1. Second, we present a quantitative evaluation of VLMs on the small-text dataset in Sec. A.2.

### A.1    Further Observations on VLMs

To validate the observations that VLMs tend to overlook detailed and insalient text instances, we conducted additional experiments with more models. For the BLIP, we calculate the ITM score which calculates the one-to-one matching probability between the image and the query. For the SigLIP models, we calculate the dot products between the vision and text embeddings without normalization following the official code. As illustrated in Fig. 4 (a), the BLIP can easily capture the salient text like "dead lakes park" but cannot find the text "old florida". However, it can capture the "old florida" when the salient regions are covered. Similar observations are presented in Fig. 4 (b) (c).

### A.2    Performance of VLMs on Small Text Dataset

To test the pre-trained ability of VLMs on fine-grained perception, we evaluate representative VLMs (CLIP [46], BLIP [18], BLIP-2 [19], SigLIP [52]) on the CTR [37] dataset. The CTR dataset is collected from the COCO dataset and features small scene text instances. As the results in Tab. 9, all of the tested VLMs struggle to retrieve images accurately. For instance, the CLIP variants obtain MAP of 8.1% while the BLIP gets 6.9% of MAP. Moreover, increasing model parameters and seen data does not lead to significant performance gains. Among these models, BLIP-2 has the highest MAP of 13.3% of the tested models, indicating limited capacity of VLMs for small text retrieval.

## B    MQTR dataset

In this section, we first supplement more construction details and annotation procedures for each subset of MQTR dataset in Sec. B.1. We then present representative samples from the MQTR dataset in Sec. B.2.

| Model | Parameters | Pre-training Data | MAP% |
|---|---|---|---|
| CLIP-RN50 | 97M | 400M images | 6.6 |
| CLIP-ViT-Base | 143M | 400M Images | 6.8 |
| CLIP-ViT-Large | 408M | 400M Images | 8.1 |
| BLIP-ViT-Large | 426M | 129M Images | 6.9 |
| BLIP-2-ViT-Large | 452M | 129M images | 13.3 |
| SigLIP-ViT-Base-512 | 194M | 9B Samples | 12.8 |
| SigLIP-ViT-Large-384 | 622M | 9B Samples | 11.7 |
| MSTAR-ViT-Base | 270M | - | 60.13 |

Table 9: Evaluation of the CLIP-style models on the CoCoText dataset. Parameters denotes the parameters of the model.

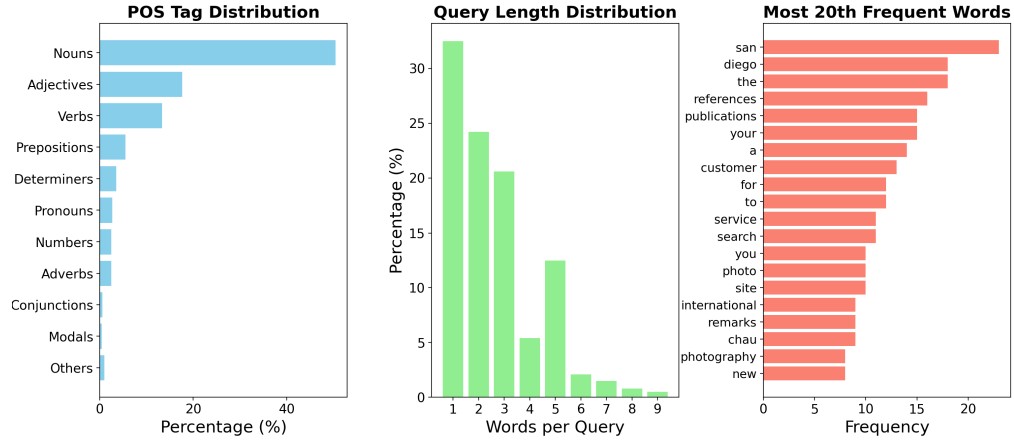

Figure 5: Statistical analysis of the MQTR benchmark.

## B.1 Construction details of the MQTR dataset

**Word Subset.** The word subset includes 5000 images and 200 word queries. The images are sourced from the test set of SVT, CTW, IC15, Total-Text and the CTR. We extract the word-level annotations from the datasets and filter out the words with less than 3 characters (e.g. "st"). After that, 200 word queries are selected according to the word frequency.

**Phrase Subset.** The images of phrase subset include 1k images from PSTR [51], 1k images manually collected from the Web. Then we use images from HierText [24]. The line-level annotations are used and the lines with only one word are filtered out. Lastly, 200 phrase queries are selected according to frequency.

**Combined Subset** We first use all images collected from the CTR and HierText dataset. Given the word and line annotations of an image, we then filter out the queries less than 3 characters. Then we implement an DFS algorithm to compute all the combinations that contain 2-4 words. Top 200 text combinations on the dataset are selected according to frequency and repeat ratios. Then images paired to these queries are first selected as positive samples. Then images containing words similar to the 200 combined queries are also selected as negative samples according to the edit distance. Lastly, there are 5000 images for the positive samples and negative samples in total.

**Semantic Subset** The semantic subset is manually collected from the web. we first brainstorm common-used scene text queries and then search for candidate images from the web. In total, the subset consists of 25 queries and 1000 images.

During the selection of multi-word queries (i.e., phrase and combined types), we manually filter out redundant entries. Specifically, queries that are overly repeated or semantically similar are removed to enhance diversity in the final set.

## B.2 Visualization of the MQTR dataset

To clarify the characteristics of the MQTR dataset, we show some examples collected from the google image search engine, as presented in Tab. 10. Since scene text typically appears as fine-grained information but the search engine often recommends the most salient images, we collect more images containing queries in fine-grained concepts. For example, "global weekly" is shown as a section name of "china daily" newspaper, adding more challenges for models. For example, for the query "dream big", the model may have difficulty distinguishing between "bream big" written on the t-shirt and "dream big" written on the shorts.

## B.3 Statistics of the MQTR dataset

In total, our dataset contains **625 unique queries** comprising **1,326 words**. As shown in Fig. 5, we report statistics including part-of-speech (POS) tag distribution, query length distribution, and the most frequent words, which collectively demonstrate the diversity of our query set across linguistic and structural dimensions.

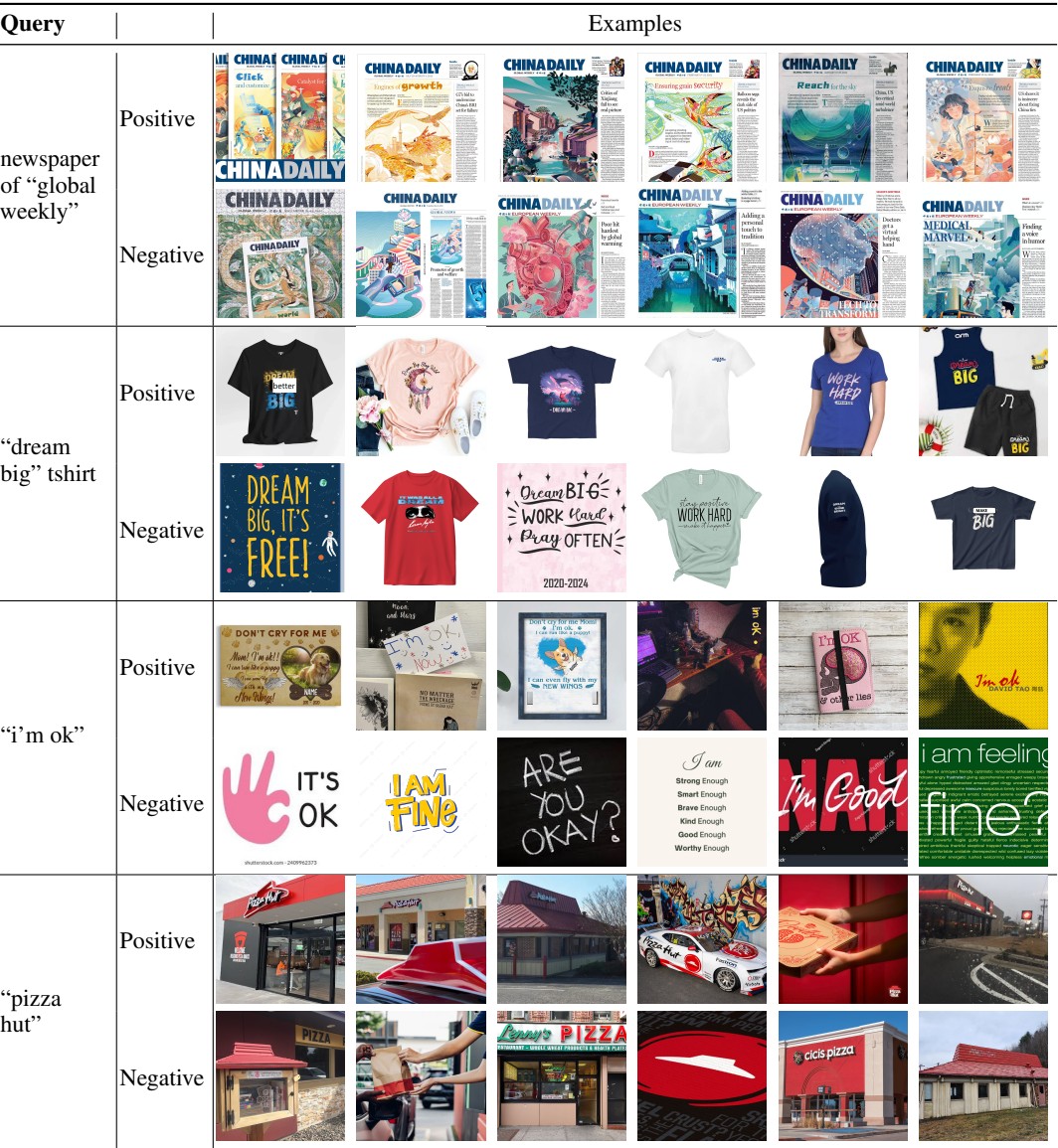

Table 10: Examples from the MQTR dataset. The Positive represents the GT images. The Negative denotes hard negative sample described in Sec. 4.

## C Experiment Details and Visualization Analysis

### C.1 Computational Efficiency Analysis.

Thanks to the design of the PVE module, image features are fed into the ViT only once. Subsequent iterations involve merely the SAS module and the Multi-modal Encoder, both of which are lightweight and thus enable a good trade-off between accuracy and efficiency. We analyze the computational cost and latency on a single A800 GPU, where the *Baseline* removes the PVE module from MSTAR.

| Method | GFLOPs | Latency (s) | FPS | CTR |
|---|---|---|---|---|
| Baseline | 248 | 0.0606 | 16.5 | 55.76 |
| MSTAR | 310 | 0.0704 | 14.2 | **60.13** |

Table 11: Computation and efficiency comparison on a single A800 GPU.

As shown in Table 11, MSTAR improves accuracy by **4.37%** on the challenging CTR dataset, while achieving an inference speed of **14.2 FPS**, over twice as fast as TG-Bridge (6.7 FPS), and maintaining comparable performance across all six benchmarks.

### C.2 Text Region Localization Analysis.

We further evaluate the localization accuracy of the predicted text regions. Binary ground-truth (GT) masks are constructed by extracting the polygon coordinates of all text regions from the CTW dataset. For comparison, we also derive binary masks from the cross-attention maps of BLIP-2 using the same processing pipeline as ours. As reported in Tab. 12, we compute the Intersection over Union (IoU) between each predicted mask and its corresponding GT mask, and report both the mean IoU and the number of high-quality masks with IoU $\geq 0.5$ across 500 images.

| Method | Mean IoU | High-Quality Masks (IoU $\geq$ 0.5) |
|---|---|---|
| BLIP-2 | 21.78 | 129 (25.8%) |
| MSTAR | **50.82** | **304 (60.8%)** |

Table 12: Quantitative comparison of text region localization on the CTW dataset. MSTAR produces substantially more accurate and higher-quality text masks than BLIP-2.

### C.3 Scale-wise Evaluation on ICDAR2015.

We further evaluate MSTAR on the ICDAR2015 dataset by analyzing performance across different text scales. For each text instance, we compute the ratio between its bounding-box area and the image area using annotations from the original datasets, and group instances into scale intervals as shown in Tab 6.

For each interval, we calculate the AP score for queries whose corresponding text instances fall within the specified scale range. For instance, given the query "apple" and a target scale range $((a, b]]$, we exclude all images where "apple" appears but its area ratio is not within that range. The AP score is then computed using the remaining valid images.

### C.4 Training Details

To ensure the reproducibility of our model, we provide the training process and the training hyper parameters, which are reported in Tab. 13. We adopt a progressive training recipe. In the first state, the model is trained on both synthetic data $D_{syn}$ and real data $D_{real}$ at an image resolution of 512. We use a learning rate of 1e-5 and linear cosine schedule with 100 warm-up steps. In this stage, we use a two-layer MLP to align the visual encoder and multi-modal encoder. In the second stage, our model is fine-tuned at the resolution of 640 only on the real data $D_{real}$. For simplicity, the hyper parameters keep the same with the first stage. In the third stage, we fine-tune the model at a higher resolution of 800 on the real data $D_{real}$. In the forth stage, the visual encoder is freezed and only the MLP, multi-modal encoder, MIM, and SAS module are optimized at a resolution of 800. For

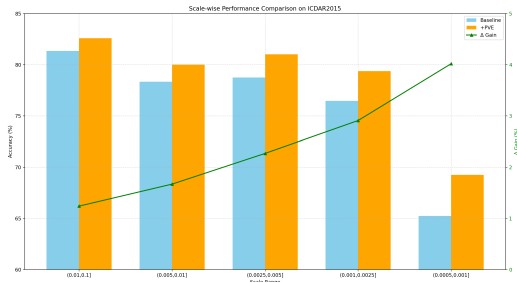

Figure 6: Scale-wise performance comparison on ICDAR2015.

|  | Phase 1 | Phase 2 | Phase 3 | Phrase 4 |
|---|---|---|---|---|
| Image Resolution | 512 | 640 | 800 | 800 |
| Learning Rate | 1e-5 | 1e-5 | 5e-6 | 5e-6 |
| Warm-Up steps | 100 | 100 | 0 | 0 |
| Freeze ViT | False | False | False | True |
| Precision of ViT | | Float | | |
| Query of $\psi$ | | 64 | | |
| RandomCrop | | True | | |
| Dataset | $\{D_{\text{syn}}, D_{\text{real}}\}$ | $D_{\text{real}}$ | $D_{\text{real}}$ | $D_{\text{real}}$ |

Table 13: Training details of our model.

the word retrieval experiment, the synthetic data $D_{\text{syn}}$ refers to 100K images randomly sampled from SynthText-900k, and the real data $D_{\text{real}}$ comes from MLT-5K. For the multi-query experiment, $D_{\text{syn}}$ includes 50K images randomly sampled from SynthText-900k and 25k images with phrase transcriptions with the synthesis engine. $D_{\text{real}}$ is the training set from the TextCap dataset. The labels are the image captions and text transcriptions acquired with Rosetta.

## C.5 Experimental Visualization Analysis

We present a qualitative analysis of the retrieval results. As illustrated in Tab. 14, our method can not only effectively leverage linguistic priors for phrase and semantic queries, but also perceive fine-grained scene text instances to achieve word and combined retrieval. For example, MSTAR successfully retrieves the combined query like "reserved" and "some" even when they appear as subtle watermarks in pictures, as depicted in the last row.

| Query | Retrieval results | | |
|---|---|---|---|
| "speed limit 25" | Bridge [11] |  | |
| | MSTAR |  | |
| "hydrate or diedrate" is written on a water bottle | Bridge [11] |  | |
| | MSTAR |  | |
| "may" | BLIP-2 (FT) [19] |  | |
| | MSTAR |  | |
| "reserved" , "some" | BLIP-2 (FT) [19] |  | |
| | MSTAR |  | |

Table 14: Qualitive analysis of the retrieval results. The green boxes mean correct samples and the red boxes denote false images. The queries and images are sampled from the MQTR dataset and CTW [21] dataset.

