# OpenReview forum: "MSTAR: Box-free Multi-query Scene Text Retrieval with Attention Recycling"
_NeurIPS.cc/2025/Conference — NeurIPS 2025 poster_

### Official Review · Reviewer_w1db · 2025-07-01

**Clarity:** 2
**Significance:** 4
**Originality:** 3
**Rating:** 4
**Confidence:** 3

**Summary:**

The paper introduces MSTAR, a box-free approach for multi-query scene text retrieval, eliminating the need for costly bounding box annotations during training.  They propose Progressive Vision Embedding to Dynamically captures multi-grained text representations by shifting attention from salient to insalient regions, and introduce Multi-Instance Matching (MIM) to enhance cross-modal alignment between vision and text embeddings. Additionally, they constract a new benchmark named MQTR with 16K images and four query types to evaluate multi-query retrieval.

**Questions:**

1.In Fig. 1 b, while it's understandable that VLMs tend to overlook detailed text instances, why do they still maintain high attention scores for regions after covering the salient text areas? I.e., after masking "Dead lakes park," the attention score remains as high as 96%.
2.In Tab. 8, why choose T=1 as the final configuration? Table 8 shows better performance at T=3, but authors chose T=1 for efficiency reasons - is this trade-off reasonable?
3.Can you provide more quantitative analysis of mask quality, such as overlap between masked regions and actual text regions?
4.Why use cross-attention for multi-word queries but Hungarian matching for single words? Is there any reason for this ?
5.The semantic query subset contains only 25 queries and 1000 images - is this scale sufficient for reliable evaluation, and how might this limited size affect the model's generalization ability on semantic queries?

**Ethical Concerns:**

["NO or VERY MINOR ethics concerns only"]

**Limitations:**

yes

**Paper Formatting Concerns:**

no comment

**Quality:**

3

**Strengths And Weaknesses:**

Strengths:
1 The proposed progressive vision embedding method is well-motivated. Leveraging the zero-shot capability of VLMs to enhance scene text retrieval tasks is very meaningful, and I think this is a mainstream trend
2 The highlight of this paper is that it addresses practical problems: high annotation cost of bounding boxes and difficulty in unified multi-query processing.
3.Well-structured paper with detailed method descriptions and helpful figures/tables for understanding

Weaknesses:
1.Still exhibits clear limitations in handling extremely small and dense text, which somewhat restricts the method's applicability
2.Despite the claimed advantages, Tables 4 and 5 reveal that MSTAR still underperforms compared to some box-based methods on specific metrics and datasets. For instance, in Table 4, MSTAR shows lower performance than TDSL on certain datasets like CTR, and in Table 5, it lags behind fine-tuned box-based methods (marked with *) such as TG-Bridge and Deepsolo on several benchmarks.
3.The MIM module design has high complexity but limited performance improvement.

---

> ### Author Rebuttal · Authors · 2025-07-31
>
> Thank you very much for your thoughtful feedback on our work!
>
> ---
>
> ***W1: Still exhibits clear limitations in handling extremely small and dense text, which somewhat restricts the method's applicability***
>
> We appreciate the reviewer's comment. While extremely small and dense text remains inherently challenging for current models, our work takes a concrete step forward within the box-free paradigm. For instance, on the CTR dataset, which features small and complex text layouts, our model achieves a MAP of 60.13%, clearly outperforming prior methods such as SPTSv2 (48.39%) and BLIP-2 (45.75%).
>
> We acknowledge that extremely small text remains an open problem for box-free method, our method brings unique advantages: it not only eliminates the box annotation cost, but also broadens the multi-query scene text retrieval applications.
>
> Therefore, we see this as a meaningful advancement in addressing fine-grained visual text understanding within box-free methods.
>
> ---
>
> ***W2: Despite the claimed advantages, Tables 4 and 5 reveal that MSTAR still underperforms compared to some box-based methods on specific metrics and datasets. For instance, in Table 4, MSTAR shows lower performance than TDSL on certain datasets like CTR, and in Table 5, it lags behind fine-tuned box-based methods (marked with \*) such as TG-Bridge and Deepsolo on several benchmarks.***
>
> For the CTR dataset that contains many extreme dense and small text instances, box-based method still hold advantages, as discussed in W1.
>
> We follow the common practice used in text spotting methods such as TG-Bridge and Deepsolo. For these methods indicated as \*, each dataset is evaluated using weights that are specifically fine-tuned on its corresponding training set. In contrast, our MSTAR model employs a single weight to evaluate all the datasets. Given that text spotting models benefit from dataset-specific fine-tuning while ours does not, a slight performance gap on these datasets is reasonable. Nevertheless, our method achieves comparable average performance against TG-Bridge (82.56 vs 82.94 ) across six datasets while offering over twice faster inference speed (14.2 FPS vs 6.7 FPS).
>
> ---
>
> ***W3: The MIM module design has high complexity but limited performance improvement.***
>
> MIM enables explicit one-to-one alignment without requiring large-scale contrastive training, making it a general and extensible component for multi-query retrieval tasks. MIM module is lightweight and introduces minimal computational overhead. MIM brings an average 1.07% improvement across all 7 benchmarks, and notably +2.19% on STR and +1.35% on CTW.
>
> ---
>
> ***Q1: In Fig. 1 b, while it's understandable that VLMs tend to overlook detailed text instances, why do they still maintain high attention scores for regions after covering the salient text areas? I.e., after masking "Dead lakes park," the attention score remains as high as 96%.***
>
> This is because the masking operation is designed with transparency, rather than completely removing the content. As shown in Fig. 1(b), although we mask out the salient text region ("Dead Lakes Park"), they remain visible. This process is similar to the design of SAS module which weakens the influence of salient regions to encourage the model to attend to overlooked fine-grained text instances. We will clarify these details in revised versions.
>
> ---
>
> ***Q2: In Tab. 8, why choose T=1 as the final configuration? Table 8 shows better performance at T=3, but authors chose T=1 for efficiency reasons - is this trade-off reasonable?***
>
> The choice of T balances effectiveness and efficiency, as shown in Table 8. Increasing T from 0 to 1 yields a significant gain (e.g., +4.37% on CTR), while further increases up to T = 3 bring smaller improvements with noticeable drops in speed. We adopt T = 1 as the final setting for its favorable trade-off between performance and inference speed.
>
> ---
>
> ***Q3: Can you provide more quantitative analysis of mask quality, such as overlap between masked regions and actual text regions?***
>
> Thank you for the suggestions! We evaluate the quality of our binary masks on the CTW dataset. During evaluation, we record the final-round binary mask used in the SAS module (T = 1). For comparison, we also obtain binary masks from the cross-attention weights of BLIP-2 using the same processing pipeline as ours.
>
> We first extract the polygon coordinates of all text regions in each image to construct a binary ground-truth (GT) mask. We then compute the Intersection over Union (IoU) between each predicted mask and its corresponding GT mask. We calculate the mean IoU across 500 images from the CTW dataset. Additionally, we count the number of masks with IoU ≥ 0.5 as high-quality masks. The experimental results are shown below.
>
> | Method | Mean IoU | High-Quality Masks (IoU ≥ 0.5) |
> |--------|----------|-------------------------------|
> | BLIP-2 | 21.78    | 129 (25.8%)                   |
> | MSTAR  | 50.82    | 304 (60.8%)                   |
>
>
> The results show that our approach generates significantly accurate text region masks compared to BLIP-2. The number of high-quality masks reaches 304 images out of all the 500 images.
>
> ---
>
> ***Q4: Why use cross-attention for multi-word queries but Hungarian matching for single words? Is there any reason for this ?***
>
> We consider each single word as an individual semantic instance and each learnable query is used to extract visual features corresponding to a single instance. This is like the DETR paradigm in object detection. Therefore Hungarian matching is sufficient and appropriate to establish one-to-one correspondence. However, the multi-word queries contain multiple semantic units. To handle this, a lightweight cross-attention layer is used to aggregate all the vision embeddings.
>
> ---
>
> ***Q5: The semantic query subset contains only 25 queries and 1000 images - is this scale sufficient for reliable evaluation, and how might this limited size affect the model's generalization ability on semantic queries?***
>
> Thanks for the comment! First, compared to previous datasets, our dataset contains a comparable number of queries and images. For example, when compared with PSTR [1], our dataset has a similar number of images (1080 vs 1000). Compared with CSVTR [2], the number of images is also close (1.6 k), and the number of queries is similar (23 vs. 25). During the collection process, we predefined a variety of query types. Data collectors were instructed to gather images from scenes such as product displays, documents, restaurants, and street views. Finally, a manual filtering step was conducted to ensure the correctness and quality of the collected data.
>
> [1] Zeng, Gangyan, et al. "Focus, distinguish, and prompt: Unleashing clip for efficient and flexible scene text retrieval." Proceedings of the 32nd ACM International Conference on Multimedia. 2024.
>
> [2] Wang, Hao, et al. "Scene text retrieval via joint text detection and similarity learning." Proceedings of the IEEE/CVF Conference on Computer Vision and Pattern Recognition. 2021.

---

### Official Review · Reviewer_5CKX · 2025-07-02

**Clarity:** 2
**Significance:** 2
**Originality:** 3
**Rating:** 3
**Confidence:** 3

**Summary:**

The paper studies the problem of scene text retrieval. To this end, the authors propose a new method named MSTAR, which eliminates bbox annotations and can handle multi-query retrieval. The proposed method consists of several components: Progressive Vision Embedding, Instruction Aware Text Representation and Muti-Instance Matching. The key insight is the Progressive Vision Embedding, which gradually shifts attention from salient features to fine-grained regions, via applying the masks derived from the cross attention map. In addition to the method, the authors have also collected a new dataset, MQTR, for the evaluation of multi-query retrieval. Experiments on existing benchmarks as well as the newly proposed benchmark show the proposed method generally achieves a better performance. Ablation study validates the effectiveness of each proposed module.

**Questions:**

Given the weaknesses of the paper, especially the significance of the problem the paper studies, I am more inclined to reject the paper.

I would suggest the authors to answer the 3 questions and concerns as in the weakness section.

**Ethical Concerns:**

["NO or VERY MINOR ethics concerns only"]

**Final Justification:**

Both of my W1 and W2 still remain unresolved (see my comment for details). Therefore, I am keeping my original rating.

**Limitations:**

Yes.

**Paper Formatting Concerns:**

No.

**Quality:**

2

**Strengths And Weaknesses:**

Strengths:

1. Dataset contribution. Compared with existing datasets, the newly constructed dataset has more images and queries, and most importantly, supports all Word, Phrase, Combined and Semantic.

2. Method contribution. The idea of Progressive Vision Embedding makes sense to me. It is an effective way of getting rid of the requirement of bbox.

Weaknesses:

1. Significance of the problem. The problem of scene text retrieval is a very specific and narrow task in the retrieval community, and may not be of interest for a large audience.

2. Performance decrease for re-ranking. In Table 5, for MSTAR, the performance degrades after re-ranking. The authors should give a reasonable explanation.

3. Paper writing. There are several writing issues: On the title, should be 'Box-Free Multi-Query'; In Line 106, should be 're-ranking'; In Line 104, should be 'BLIP-2'; In Table 5 caption, should be 'fine-tuned'. The paper needs further proofreading and attention to make the wordings and spellings correct and professional.

---

> ### Author Rebuttal · Authors · 2025-07-31
>
> Thank you very much for your thoughtful feedback on our work!
>
> ---
>
> ***W1: Significance of the problem. The problem of scene text retrieval is a very specific and narrow task in the retrieval community, and may not be of interest for a large audience.***
>
> We would like to clarify that scene text retrieval is a fundamental task in the fields of document analysis and OCR. It has attracted significant attention from the research community, with a number of influential works published in top-tier venues such as TPAMI [1], CVPR [2,7], ECCV [5], ICML [8], and ACM MM [4].
>
> Moreover, scene text serves as an important cue in information retrieval. Incorporating scene text information has been shown to improve the performance of general image-text matching tasks on benchmarks such as MS-COCO and Flickr30K [3,6].
>
> Finally, due to the inherently fine-grained and intricate nature of scene text, we believe it presents unique challenges that warrant further investigation. Our work proposes a meaningful box-free method for unified multi-query scene text retrieval (Reviewer w1db). The PVE is well-motivated and interesting for capturing scene text at different scales (Reviewer EPGg, w1db). The newly introduced MQTR dataset enables thorough and realistic evaluation of retrieval models (Reviewer Qggy, EPGg). To sum up, we argue that scene text retrieval is not only necessary but also a valuable direction for the NeurIPS community.
>
> [1] Wang, Hao, et al. "Partial scene text retrieval." IEEE Transactions on Pattern Analysis and Machine Intelligence (2024).
>
> [2] Wang, Hao, et al. "Scene text retrieval via joint text detection and similarity learning." Proceedings of the IEEE/CVF Conference on Computer Vision and Pattern Recognition. 2021.
>
> [3] Cheng, Mengjun, et al. "Vista: Vision and scene text aggregation for cross-modal retrieval." Proceedings of the IEEE/CVF Conference on Computer Vision and Pattern Recognition. 2022.
>
> [4] Zeng, Gangyan, et al. "Focus, distinguish, and prompt: Unleashing clip for efficient and flexible scene text retrieval." Proceedings of the 32nd ACM International Conference on Multimedia. 2024.
>
> [5] Gómez, Lluís, et al. "Single shot scene text retrieval." Proceedings of the European conference on computer vision (ECCV). 2018.
>
> [6] Mafla, Andrés, et al. "Stacmr: Scene-text aware cross-modal retrieval." Proceedings of the IEEE/CVF Winter Conference on Applications of Computer Vision. 2021.
>
> [7] Qin, Xugong, et al. "CLIP is Almost All You Need: Towards Parameter-Efficient Scene Text Retrieval without OCR." Proceedings of the Computer Vision and Pattern Recognition Conference. 2025.
>
> [8] Li, Gengluo, Huawen Shen, and Yu Zhou. "Beyond Cropped Regions: New Benchmark and Corresponding Baseline for Chinese Scene Text Retrieval in Diverse Layouts." arXiv preprint arXiv:2506.04999 (2025).
>
> ---
>
>
> ***W2: Performance decrease for re-ranking. In Table 5, for MSTAR, the performance degrades after re-ranking. The authors should give a reasonable explanation.***
>
> In Table 5, we observe the performance improves by 1.62% on average after re-ranking. However, the results slightly degrade on the SVT (91.31% vs 91.11%) and STR (86.25% vs 86.14%) datasets. This is because the re-ranking is an additional component to refine the top-k results. Re-ranking can only reorder a fixed set of top-k results. If the top-k set already contains false positives (e.g., visually similar but semantically irrelevant items), the reranker may overfit to superficial cues (e.g., layout or text frequency) and push wrong candidates higher, thereby amplifying early-stage errors.
>
> ---
>
> ***W3: Paper writing. There are several writing issues: On the title, should be 'Box-Free Multi-Query'; In Line 106, should be 're-ranking'; In Line 104, should be 'BLIP-2'; In Table 5 caption, should be 'fine-tuned'. The paper needs further proofreading and attention to make the wordings and spellings correct and professional.***
>
> Thank you for pointing this out! We will correct all the mentioned issues and perform thorough proofreading to ensure that the paper meets the professional standards in wording, spelling, and overall clarity.

---

> ### Author Response · Authors · 2025-08-06
>
> Dear Reviewer 5CKX,
>
> Thank you for your time and valuable feedback. As the discussion period will be ending on August 8th, we remain open and willing to address any remaining questions or concerns you may have. We would greatly appreciate it if you could consider improving the evaluation after reviewing our responses. Thank you very much for your consideration.
>
> Sincerely,
>
> Paper 20746  Authors

---

> ### Comment · Reviewer_5CKX · 2025-08-06
>
> Thank the authors for rebuttal.
>
> For W1, in my opinion I still respectfully disagree that the problem of scene text retrieval is as significant as general t2i retrieval; given the more and more powerful models these days, the problem of OCR has been largely solved. I don't think the topic of the paper will be of interest of a large audience at NeurIPS.
>
> For W2, although the authors have made some explanations, the fact is the proposed method for re-ranking is not robust. Therefore, my concern has not been addressed.
>
> Therefore, I am keeping my original rating.

---

> > ### Author Response · Authors · 2025-08-07
> >
> > First and foremost, we would like to thank you for your constructive and insightful comments. Through previous discussion, we believe we have reached some consensus, i.e, the significance of scene text retrieval in the OCR domain. To further clarify our perspective, we would like to elaborate on two key points below:
> >
> > ---
> >
> > W1: The significance of scene text retrieval.
> >
> >
> > - **First, although models are becoming more powerful, OCR remains unsolved, particularly in the context of fine-grained scene text understanding.** For example, Gemini 2.5 Pro, one of the most advanced MLLMs to date, achieves only 59.3 on OCRBench v2 \[1]. CC-OCR \[2] further concludes that fine-grained text grounding is a common limitation across all current models.
> >
> > - **Second, general image-text retrieval models cannot handle the unique challenges of scene text retrieval.** As shown in the table below, state-of-the-art text-to-image retrievers achieve less than 14% MAP on the CTR dataset. This result highlights the distinct nature of scene text retrieval and the necessity for specialized solutions. Our MSTAR, achieves 60.13%, clearly demonstrating advantages in this domain.
> >
> > - **Third, scene text retrieval can facilitate fine-grained visual representation applications, given its object-level characteristics.** Recent studies have shown that general retrieval models struggle with fine-grained and instance-level matching [3, 4]. Scene text retrieval, as a fundamental embedding task that focuses on OCR instances, can facilitate downstream tasks such as text rendering [5] and document retrieval [6], which require fine-grained visual information. For example, [7] finds that capturing detailed scene text features is a critical bottleneck in text generation quality.
> >
> >
> > | Model                | Venue   | CTR (MAP%) |
> > | -------------------- | ------- | ---------- |
> > | CLIP-ViT-Large       | ICML'21 | 8.1        |
> > | BLIP-ViT-Large       | ICML'22 | 6.9        |
> > | BLIP2-ViT-Large      | ICML'23 | 13.3       |
> > | SigLIP-ViT-Large-384 | ICCV'23 | 11.7       |
> > | FG-CLIP-Large        | ICML'25 | 10.9       |
> > | **MSTAR-ViT-Base**   | **-**    | **60.13**  |
> >
> > [1] Fu, Ling, et al. "Ocrbench v2: An improved benchmark for evaluating large multimodal models on visual text localization and reasoning." arXiv preprint arXiv:2501.00321 (2024).
> >
> > [2] Yang, Zhibo, et al. "Cc-ocr: A comprehensive and challenging ocr benchmark for evaluating large multimodal models in literacy." arXiv preprint arXiv:2412.02210 (2024).
> >
> > [3] Jing, Dong, et al. "Fineclip: Self-distilled region-based clip for better fine-grained understanding." Advances in Neural Information Processing Systems 37 (2024): 27896-27918.
> >
> > [4] Xie, Chunyu, et al. "FG-CLIP: Fine-grained visual and textual alignment." Proceedings of the 40th International Conference on Machine Learning. 2025.
> >
> > [5] Chen, Jingye, et al. "Textdiffuser-2: Unleashing the power of language models for text rendering." European Conference on Computer Vision. Cham: Springer Nature Switzerland, 2024.
> >
> > [6] Faysse, Manuel, et al. "Colpali: Efficient document retrieval with vision language models." arXiv preprint arXiv:2407.01449 (2024).
> >
> > [7] Wang, Alex Jinpeng, et al. "Beyond Words: Advancing Long-Text Image Generation via Multimodal Autoregressive Models." arXiv preprint arXiv:2503.20198 (2025).
> >
> > ---
> >
> > W2: The re-ranking process.
> >
> > As discussed in the rebuttal, we suspect that the slight performance drops on SVT (−0.2%) and STR (−0.11%) because only the top 2% images were re-ranked. To support this, we increase the number of re-ranked images. As shown in the tables below, performance improves as more images are re-ranked, confirming the effectiveness of this module. Moreover, we emphasize that the re-ranking module brings significant gains on other datasets, i.e, a +5.12% improvement on the CTR dataset.
> >
> > *Re-ranking performance on the SVT (249 images) and STR (10,000 images) dataset.* *\* denotes the results reported in the submission.*
> >
> > | Re-ranked image number | 0     | 5 (2%)   | 20 (8%)            | 40  (16%)           |
> > | --------------------- | ----- | ----- | -------------- | -------------- |
> > | SVT (MAP %)       | 91.31 | 91.11\* | 91.48 (+0.17%) | 92.08 (+0.77%) |
> >
> > |  Re-ranked image number  | 0     | 200 (2%)  | 400  (4%)          | 800  (8%)         |
> > | --------------------- | ----- | ------ | -------------- | -------------- |
> > | STR (MAP %)      | 86.25 | 86.14\*  | 86.88 (+0.63%) | 87.39 (+1.14%) |
> >
> > ---
> >
> > We hope that our responses have sufficiently addressed your concerns. If you still have any remaining questions or suggestions, please do not hesitate to let us know. We would be more than willing to provide further discussion or clarification as needed.

---

> > > ### Comment · Reviewer_5CKX · 2025-08-08
> > >
> > > Thank the authors for their reply.
> > >
> > > For W1, I am still not fully convinced that the task can benefit a wide range of applications as the more general t2i retrieval does. Therefore, I am not sure about whether it would be suitable for NeurIPS.
> > >
> > > For W2, although the model can reach a higher re-ranking performance after increasing he re-ranked image number, the fact that it degrades the performance at a lower re-ranked image number is an evidence the method is not robust. In practice, the re-ranked image number cannot be a parameter that we can tune. Rather, it should be fixed, and the evidence shows that the method cannot give a consistent improvement.

---

> ### Author Response · Authors · 2025-08-08
>
> We sincerely appreciate the reviewers for openly and thoroughly raising their concerns. In response to the aforementioned questions, we provide the following further discussion:
>
> ---
>
> W1: The significance of the task.
>
> - **Applications of scene text retrieval.** As a fundamental OCR task, scene text retrieval can be directly applied to video frame extraction [1] and library book retrieval [2]. It also benefits related tasks such as text rendering, general cross-modal retrieval, and visual document retrieval (discussed in the last reply).
>
> - **Specialized retrieval tasks published at NeurIPS.** Although scene text retrieval may not have as broad a range of downstream applications as general text-to-image methods, it stands out as a unique object-level retrieval task with distinct challenges, as discussed earlier. Numerous specialized retrieval tasks have been published at NeurIPS, including person re-identification [3], signature retrieval [4], fashion retrieval [5], and scientific document retrieval [6]. Therefore, we believe that acceptance of our paper will undoubtedly inspire further research in this area.
>
> [1] Yang, Xiao, et al. "Smart library: Identifying books on library shelves using supervised deep learning for scene text reading." 2017 ACM/IEEE Joint Conference on Digital Libraries (JCDL). IEEE, 2017.
>
> [2] Song, Hao, et al. "Text siamese network for video textual keyframe detection." 2019 International Conference on Document Analysis and Recognition (ICDAR). IEEE, 2019.
>
> [3] Eom, Chanho, and Bumsub Ham. "Learning disentangled representation for robust person re-identification." Advances in neural information processing systems 32 (2019).
>
> [4] Qi, Daiqing, Handong Zhao, and Sheng Li. "Easy Regional Contrastive Learning of Expressive Fashion Representations." Advances in Neural Information Processing Systems 37 (2024): 20480-20509.
>
> [5] Zhang, Peirong, et al. "Msds: A large-scale chinese signature and token digit string dataset for handwriting verification." Advances in Neural Information Processing Systems 35 (2022): 36507-36519.
>
> [6] Wang, Jianyou Andre, et al. "Scientific document retrieval using multi-level aspect-based queries." Advances in Neural Information Processing Systems 36 (2023): 38404-38419.
>
> ---
>
> W2: The effectiveness of the re-ranking module.
>
> - First, as shown in Tab. 4 and Tab. 5, without re-ranking, the performance of our MSTAR achieves comparable performance of box-free methods while eliminates the costly box supervision.
>
> - Secondly, re-ranking is not always required to outperform coarse ranking especially the inital performance is high. For instance, [1] found that bge-reranker-v2-m3 can degrade performance when the initial ranker voyage-2 is very accurate. [2] also found re-ranking degrads on the TriviaQA dataset. Since the inital MAP of SVT (91.31%) and STR (86.25%) is high, the top 2% candidate images are already high-quality, so it is reasonable that re-ranking may introduce some noise.
>
> - Thirdly, experimental results have shown the overall improvement on the re-ranking module (Tab. 4) and the effectiveness on STR and SVT when image number is set to relative large.
>
> [1] Jacob, Mathew, et al. "Drowning in documents: consequences of scaling reranker inference." arXiv preprint arXiv:2411.11767 (2024).
>
> [2] Tang, Qiaoyu, et al. "Self-retrieval: End-to-end information retrieval with one large language model." Advances in Neural Information Processing Systems 37 (2024): 63510-63533.
>
> ---
> We appreciate your thoughtful feedback and hope our replies have clarified your concerns. Should you have any further questions or wish to discuss anything else, we are always ready to assist and offer any additional clarification you may require.

---

> > ### Author Response · Authors · 2025-08-09
> >
> > Dear Reviewer 5CKX,
> >
> > We truly appreciate your insightful comments and suggestions. With the discussion period coming to an end, please let us know if you have any additional questions.  We will respond promptly. Thanks for your consideration.
> >
> > Respectfully,
> >
> > Paper 20746 Authors

---

> ### Author Response · Authors · 2025-08-09
> **Summary of our constructive discussions with Reviewer 5CKX so far**
>
> We summarize our discussion as follows:
>
> **W1. Significance of scene text retrieval**
>
> * This task is important both for OCR and general image–text matching. Numerous works of this domain in TPAMI, ICML, and CVPR, etc., highlight its significance.
> * Our evidence shows that both OCR and scene text retrieval remain challenging for state-of-the-art MLLMs and general retrievers.
> * Scene text retrieval is a unique, object-level cross-modal retrieval task with wide applications and strong parallels to other specialized tasks (e.g., person re-ID) accepted at NeurIPS.
>
> **W2. Effectiveness of the re-ranking module**
>
> * Without re-ranking, MSTAR is comparable to box-based methods while removing costly box supervision.
> * Re-ranking underperforms only when initial performance is already high and the re-ranked top-K is small, which is reasonable as other re-rankers do.
> * Experiments show consistent gains on overall improvement and on STR and SVT with larger image numbers.
>
> ---
> We believe we have fully addressed all concerns raised so far. If there are any remaining questions, please reach out soon, as the discussion period is nearing its end. We are happy to provide any further clarification. We are always here to help. Many thanks!

---

### Official Review · Reviewer_EPGg · 2025-07-05

**Clarity:** 3
**Significance:** 3
**Originality:** 3
**Rating:** 4
**Confidence:** 3

**Summary:**

A box-free scene-text retreival method for multiple query styles is proposed in this manuscript. To achieve box-free retreival, this method utilizes saliency map from vision encoder for text spotting. An attetion recycling mechanism is proposed in this method for saliency attention shift in order to capture the text from saliency part to less saliency part. An instruction aware text representation is introduced for retrieval with free-style query. The instruction indicates the query type, and queries of different type are mapped to their own feature space during trianing. A multi-instance matching method is involved to match vision features under different attention shift with text queries. The proposed method is evaluated on multiple benchmark datasets and a newly collected dataset for free-style scene-text retreival task. The experiment results show the effectiveness of the proposed method.

**Questions:**

1.	In Eq.2, is the mask directly applied on the feature map or it needs a mapping or it is an attention map?

**Ethical Concerns:**

["NO or VERY MINOR ethics concerns only"]

**Final Justification:**

My concerns have been well-addressed. The idea of progressive attention can not only be applied in small-scale text retrieval, but be extended to discover small-scale objects as well. Therefore the idea proposed in this paper is valuable. After reading the rebuttal and the reviews from other reviewers, I tend to keep my rating as borderline accept.

**Limitations:**

Yes.

**Paper Formatting Concerns:**

No concern.

**Quality:**

3

**Strengths And Weaknesses:**

**Strengths:**

1.	The method achieves performance comparable to the state-of-the-arts not only on scene text retrieval task but also on scene text spotting task;

2.	The introducing of attention shift for captureing scene-text of different scales is interesting, and help achieve good performance;

3.	A new dataset is collected for free-style scene text retreival task;

4.	The propsoed method is extensively evaluated;

**Weakness:**

1.	The authors claim that the proposed method is able to process text queries of different styles, and an instruction aware text representation is proposed to well process the text queries of four styles. This instruction-based way may not be pratical in real world application as the users tend to input queries without indicating their styles. An addiional query style recognition will make this work more complete.

2.	Vision Encoder like Sig-LIP may not be suitable for scene text saliency detection as the model is pre-trained on data most of which do not have scene text in image. The saliency may appear at the apearance of the object the corresponding text describe, the appeat at the scene text. The finetuning needs large scale of data to calibrate the vision encoder for scene text saliency adjustment;

3.	The authors state that the attention recycling can help discover scene text with less saliency. However, an evaluation specifically designed for small-scale scene text is not conducted. It would be great to also include this experiment to make the evaluation complete. The evaluation of scene text retreival performance under different text scale is necessary;

4.	The attention shift (or progressive enhancement of feature) is a popular strategy in computer vision, which has been used in other vision task such as object saliency detection [1], action recognition [2], action prediction [3], image captioning [4]. It would be great to also review these works in related work;

5.	Typos:

* In line 153, the sentence “The N is the total number of text queries paired to the image.” is repeated twice;
* In the caption of Table 3. The word “phrase-level”  is wrongly written as “Prase-level”;

[1] Siris, Avishek, Jianbo Jiao, Gary KL Tam, Xianghua Xie, and Rynson WH Lau. "Inferring attention shift ranks of objects for image saliency." In Proceedings of the IEEE/CVF conference on computer vision and pattern recognition, pp. 12133-12143. 2020.

[2] Weng, Junwu, Donghao Luo, Yabiao Wang, Ying Tai, Chengjie Wang, Jilin Li, Feiyue Huang, Xudong Jiang, and Junsong Yuan. "Temporal distinct representation learning for action recognition." In European Conference on Computer Vision, pp. 363-378. Cham: Springer International Publishing, 2020.

[3] Stergiou, Alexandros, and Dima Damen. "The wisdom of crowds: Temporal progressive attention for early action prediction." In Proceedings of the IEEE/CVF Conference on Computer Vision and Pattern Recognition, pp. 14709-14719. 2023.

[4] Pedersoli, Marco, Thomas Lucas, Cordelia Schmid, and Jakob Verbeek. "Areas of attention for image captioning." In Proceedings of the IEEE international conference on computer vision, pp. 1242-1250. 2017.

---

> ### Author Rebuttal · Authors · 2025-07-31
>
> Thank you very much for your thoughtful feedback on our work!
>
> ---
>
> ***W1: The authors claim that the proposed method is able to process text queries of different styles, and an instruction aware text representation is proposed to well process the text queries of four styles. This instruction-based way may not be pratical in real world application as the users tend to input queries without indicating their styles. An addiional query style recognition will make this work more complete.***
>
> Thanks for your valuable suggestion! To enhance the practicality of our method in real-world scenarios where users often do not explicitly indicate query styles, we introduced a rule-based query style classification module.
>
> Since OCR text and general visual concepts can be confusing, users are asked to put OCR-related text inside single quotes. First, we apply NLTK tokenizer to segment the input query into tokens. Based on the tokenized results and simple pattern matching, we classify queries into four categories as follows:
>
> - **Word queries:** The query consists of a single quoted token, e.g., `'apple'`. When the tokenized query contains exactly one word enclosed in single quotes, it is recognized as a word query.
> - **Phrase queries:** The query is a single quoted sequence of multiple continuous words forming a meaningful phrase, e.g., `'scene text retrieval'`. If the quoted text contains more than one word without any spaces separating different quoted segments, it is identified as a phrase query.
> - **Combined queries:** The query contains multiple quoted words or phrases separated by spaces, e.g., `'retrieval' 'OCR'`. If the input has multiple quoted segments separated by spaces, it is classified as a combined query.
> - **Semantic queries:** Queries that have a more free-form structure, possibly mixing quoted and unquoted parts, such as `bottle saying 'purified drinking water'`. These are identified when the query does not strictly follow the above patterns.
>
> This approach leverages simple regular expressions over tokenized input, providing a lightweight and robust classification mechanism without dependence on heavy NLP models. We believe this addition improves the completeness and applicability of our framework significantly.
>
> ---
>
> ***W2: Vision Encoder like Sig-LIP may not be suitable for scene text saliency detection as the model is pre-trained on data most of which do not have scene text in image. The saliency may appear at the apearance of the object the corresponding text describe, the appeat at the scene text. The finetuning needs large scale of data to calibrate the vision encoder for scene text saliency adjustment;***
>
> Thanks for your valuable suggestion! The reasons for choosing SigLIP as the vision encoder are two-fold:
>
> First, vision encoders such as SigLIP exhibit strong vision-language alignment capabilities, as they are pretrained on massive web image–caption pairs, which often include rich scene text information [1,2].
>
> Second, SigLIP and similar encoders demonstrate impressive zero-shot performance on scene text understanding tasks. For instance, TCM [3]/TCM++ [4] shows that neurons in CLIP are inherently responsive to textual signals, and CLIP can enhance text spotting models with significantly reduced supervision. Additionally, FDP [5] reports that CLIP achieves strong zero-shot performance on the SVT dataset, with an accuracy of 65.07%, further validating its effectiveness in handling scene text without task-specific fine-tuning.
>
> [1] Fan, David, et al. "Scaling language-free visual representation learning." arXiv preprint arXiv:2504.01017 (2025).
>
> [2] Cao, Liangliang, et al. "Less is more: Removing text-regions improves clip training efficiency and robustness." arXiv preprint arXiv:2305.05095 (2023).
>
> [3] Yu, Wenwen, et al. "Turning a clip model into a scene text detector." Proceedings of the IEEE/CVF conference on computer vision and pattern recognition. 2023.
>
> [4] Yu, Wenwen, et al. "Turning a clip model into a scene text spotter." IEEE Transactions on Pattern Analysis and Machine Intelligence 46.9 (2024): 6040-6054.
>
> [5] Zeng, Gangyan, et al. "Focus, distinguish, and prompt: Unleashing clip for efficient and flexible scene text retrieval." Proceedings of the 32nd ACM International Conference on Multimedia. 2024.
>
> ---
>
> ***W3: The authors state that the attention recycling can help discover scene text with less saliency. However, an evaluation specifically designed for small-scale scene text is not conducted. It would be great to also include this experiment to make the evaluation complete. The evaluation of scene text retreival performance under different text scale is necessary;***
>
> We conduct evaluation experiments on the ICDAR2015 dataset by analyzing performance across different text scales. Specifically, we compute the ratio of each text instance’s area to the corresponding image area based on the dataset annotations, and then group these instances into scale intervals as shown in the table below.
>
> To evaluate retrieval performance at different scales, we calculate the AP score for each query using only the images that contain matching text instances falling within the specified scale range. For example, given a query "apple" and a target scale range \((a, b]\), we discard any images where the word "apple" appears but its occupied area ratio is not within \((a, b]\). The AP score is then computed over the remaining valid images.
>
>
> | **Scale Range** | **Baseline** | **+PVE** | **Δ**  |
> | --------------- | ------------ | -------- | ------ |
> | (0.01, 0.1]     | 81.35        | 82.59    | +1.24% |
> | (0.005, 0.01]   | 78.34        | 80.01    | +1.67% |
> | (0.0025, 0.005] | 78.75        | 81.02    | +2.27% |
> | (0.001, 0.0025] | 76.47        | 79.38    | +2.91% |
> | (0.0005, 0.001] | 65.23        | 69.25    | +4.02% |
> | (0, 1)          | 79.23        | 81.21    | +1.98% |
>
> The first column indicates the text scale intervals, the second shows the baseline performance without PVE, the third presents the results with our proposed PVE module, and the last column reports the performance gain. As shown, the smaller the text instance, the more significant the improvement brought by PVE. This confirms that our method is particularly effective at capturing fine-grained text, leading to larger gains in challenging small-scale scenarios.
>
>
> ---
>
> ***W4: The attention shift (or progressive enhancement of feature) is a popular strategy in computer vision, which has been used in other vision task such as object saliency detection [1], action recognition [2], action prediction [3], image captioning [4]. It would be great to also review these works in related work***
>
> Thanks for the suggestions, we will add these related works in the revised versions.
>
> ---
>
> ***W5: Typos: In line 153, the sentence “The N is the total number of text queries paired to the image.” is repeated twice; In the caption of Table 3. The word “phrase-level” is wrongly written as “Prase-level”;***
>
> Thanks for pointing this out. We will correct the repeated sentence in line 153 and the typo “Prase-level” in the caption of Table 3 in the revised version.
>
> ---
>
> ***Q1: In Eq.2, is the mask directly applied on the feature map or it needs a mapping or it is an attention map?***
>
> The mask is directly add to the attention weights like Mask2Former. Here's a concise explanation of Eq.2.
>
> Given the input image features $f_{t-1} \in \mathbb{R}^{B \times L \times D}$, the model computes:
>
> $$
> Q = f_{t-1}W_Q,\quad K = f_{t-1}W_K,\quad V = f_{t-1}W_V,
> $$
>
> where $Q, K, V \in \mathbb{R}^{B \times L \times D}$ are the projected query, key, and value embeddings. Then, the self-attention scores are calculated by:
>
> $$
> \text{Scores} = \frac{QK^\top}{\sqrt{d}}.
> $$
>
> To suppress salient (already attended) regions and enhance less-attended areas, we apply the **binary mask** **$M_{t-1} \in \mathbb{R}^{B \times L}$**, which is derived from the inverted cross-attention map. This mask is broadcast and added to the attention scores:
>
> $$
> \text{Scores} = \text{Scores} + M_{\text{expanded}},
> $$
>
> where masked positions (i.e., previously over-attended regions) receive large negative values, effectively reducing their contribution via:
>
> $$
> \text{Probs} = \text{softmax}(\text{Scores}).
> $$
>
> Finally, the new features are computed as:
>
> $$
> f_t = \text{Probs} \cdot V.
> $$
>
> Through this masked attention, the SAS module adaptively shifts focus away from dominant regions and encourages the model to attend to previously overlooked text instances.
>
> [1] Cheng, Bowen, et al. "Masked-attention mask transformer for universal image segmentation." Proceedings of the IEEE/CVF conference on computer vision and pattern recognition. 2022.

---

> > ### Comment · Reviewer_EPGg · 2025-08-05
> >
> > Thanks for the detailed response from the authors. I have no further concern and decide to keep my rate as Borderline accept.

---

### Official Review · Reviewer_Qggy · 2025-07-05

**Clarity:** 3
**Significance:** 3
**Originality:** 3
**Rating:** 4
**Confidence:** 4

**Summary:**

This work proposes a box-free scene text retrieval method that eliminates the need for bounding box annotations. It employs progressive vision embedding and style-aware queries to enhance text representation and vision-language alignment. Additionally, it introduces MQTR, a new dataset for multi-query retrieval, on which the method outperforms existing approaches.

**Questions:**

1. Regarding the new MQTR dataset, the selection of 200 queries for the word, phrase, and combined subsets was based on frequency from a collection of existing datasets. How did you ensure these queries are diverse and representative, and not biased towards the specific text distributions of the source datasets (e.g., SynthText, TextCap)?
2. The paper identifies difficulty with "extremely small text" as a primary limitation. Why donot your method deal with this limitation?
3. How to set proper recurrent steps T?

**Ethical Concerns:**

["NO or VERY MINOR ethics concerns only"]

**Limitations:**

Yes

**Quality:**

2

**Strengths And Weaknesses:**

Strengths:

1. This paper proposes a new box-free method specifically designed for multi-query scene text retrieval. This work may benifit ***
2. The new dataset MQTR dataset enables a more thorough and realistic evaluation of retrieval models and will likely spur further research in this domain.
3. MSTAR demonstrates impressive performance gains across seven public datasets and the new MQTR benchmark.


Weaknesses:
1. The model's performance deteriorates on datasets with extremely small and dense text, where box-based methods still hold an advantage.
2. Its core attention recycling mechanism introduces a direct trade-off between higher retrieval accuracy and slower processing speed.
3. The algorithm for shifting the model's attention is complex, potentially posing a challenge for replication and future development.

---

> ### Author Rebuttal · Authors · 2025-07-31
>
> Thank you very much for your thoughtful feedback on our work! Because both W1 and Q2 ask about the "extremely small text", we answer the two questions together.
>
> ---
>
> ***W1: The model's performance deteriorates on datasets with extremely small and dense text, where box-based methods still hold an advantage.***
>
> ***Q2: The paper identifies difficulty with "extremely small text" as a primary limitation. Why does your method not deal with this limitation?***
>
> Thank you for the valuable comment. Handling extremely small text remains inherently challenging for all box-free methods without precise box supervision. Nonetheless, our work takes a concrete step forward within the box-free paradigm. For instance, on the CTR dataset, which features small and complex text layouts, our model achieves a MAP of 60.13%, clearly outperforming prior methods such as SPTSv2 (48.39%) and BLIP-2 (45.75%).
>
> While extremely small text remains an open problem for box-free methods, our method brings unique advantages: it not only eliminates the box annotation cost, but also broadens the multi-query scene text retrieval applications.
>
> Therefore, we see this as a meaningful advancement in addressing fine-grained visual text understanding within box-free methods.
>
> ---
>
> ***W2: Its core attention recycling mechanism introduces a direct trade-off between higher retrieval accuracy and slower processing speed.***
>
> Thank you for the comment. Due to the design of our PVE module, the image features are only fed into the ViT once. Subsequent iterations involve only the SAS module and the Multi-modal Encoder. These two components are relatively lightweight in terms of computation time, enabling our method to achieve a favorable balance between accuracy and efficiency.
>
> To study the computation costs of each component, we provide the analysis of the computation and latency of MSTAR. The baseline refers to the results obtained by removing the PVE module from MSTAR. The experiments are conducted on a single A800 GPU.
>
> | Method   | Computation (GFLOPs) | Latency (s) | FPS  | CTR    |
> |----------|----------------------|-------------|------|--------|
> | Baseline | 248                  | 0.0606      | 16.5 | 55.76  |
> | MSTAR    | 310                  | 0.0704      | 14.2 | 60.13  |
>
> As the results show, our method improves precision by 4.37% on the challenging CTR dataset. It also achieves an inference speed of 14.2 FPS, which is more than twice as fast as TG-Bridge (6.7 FPS), while maintaining comparable performance across the six datasets.
>
> ---
>
> ***W3: The algorithm for shifting the model's attention is complex, potentially posing a challenge for replication and future development.***
>
> Thank you for the comment. In our design, the PVE module performs attention shifting in an effective (Reviewer 5CKX) and interesting (Reviewer EPGg) manner. It directly reuses the cross-attention weights from the model as binary mask to shift the focus toward less salient features. Notably, our approach eliminates the complex pipeline adopted by previous box-based methods, which typically require explicit scene text detection [1] and ROI-based embedding [2]. This further reducing implementation complexity.
>
> Regarding future development, all components in our design are modular and plug-and-play. The attention shift process can be iteratively controlled via a hyperparameter T. They can be flexibly integrated with different types of text queries and various visual backbones. We believe this work can inspire further research and benefit the community.
>
> Regarding reproducibility, we have included all training details, code, and pretrained weights in the supplementary material. We will release them publicly to the open-source community. These resources ensure the reproducibility of our results.
>
> [1] Huang, Mingxin, et al. "Bridging the gap between end-to-end and two-step text spotting." Proceedings of the IEEE/CVF Conference on Computer Vision and Pattern Recognition. 2024.
>
> [2] Wang, Hao, et al. "Partial scene text retrieval." IEEE Transactions on Pattern Analysis and Machine Intelligence (2024).
>
> ---
>
> ***Q1: Regarding the new MQTR dataset, the selection of 200 queries for the word, phrase, and combined subsets was based on frequency from a collection of existing datasets. How did you ensure these queries are diverse and representative, and not biased towards the specific text distributions of the source datasets (e.g., SynthText, TextCap)?***
>
> We demonstrate the diversity and representativeness of our queries from three perspectives:
>
> First, our source data is collected from 8 widely used datasets, which enables diversity of both queries and images. These datasets cover a wide range of text types, including single-word text, curved text (CTW), line-level text (CTW, HierText), small text (CTR), and dense text (HierText), among others. In addition, the images and annotations used in MQTR are entirely independent of our training datasets, including SynthText and TextCap.
>
> Second, during the selection process for multi-word queries (i.e., phrase and combined types), we manually filtered out entries with high redundancy. For example, queries that were overly repeated or semantically similar were removed to further ensure diversity in the final query set.
>
> Third, we provide a detailed statistical analysis of the queries. In total, there are 625 unique queries comprising 1,326 words. We report statistics including part-of-speech (POS) tag distribution, query length distribution, character frequency, and word frequency. These analyses demonstrate the diversity of our queries across multiple linguistic and structural dimensions.
>
> 1. POS Tag Distribution
>
> | POS          | Percentage (%) |
> | ------------ | -------------- |
> | Nouns        | 50.45          |
> | Adjectives   | 17.65          |
> | Verbs        | 13.35          |
> | Prepositions | 5.51           |
> | Determiners  | 3.54           |
> | Pronouns     | 2.64           |
> | Numbers      | 2.49           |
> | Adverbs      | 2.49           |
> | Conjunctions | 0.53           |
> | Modals       | 0.38           |
> | Others       | 0.98           |
>
> 2. Query Length Distribution
>
> | Words per Query | Percentage (%) |
> | --------------- | -------------- |
> | 1 word          | 32.5           |
> | 2 words         | 24.2           |
> | 3 words         | 20.6           |
> | 4 words         | 5.4            |
> | 5 words         | 12.5           |
> | 6 words         | 2.1            |
> | 7 words         | 1.5            |
> | 8 words         | 0.8            |
> | 9 words         | 0.5            |
>
> 3. Most Frequent Words (Top 20):
>
> ```
> san (23), diego (18), the (18), references (16), publications (15),
> your (15), a (14), customer (13), for (12), to (12),
> service (11), search (11), you (10), photo (10), site (10),
> international (9), remarks (9), chau (9), photography (8), new (8)
> ```
> ---
>
> ***Q3: How to set proper recurrent steps T?***
>
> The choice of T should balance effectiveness and efficiency. We report the ablation results of T in Table 8. As T increases from 0 to 1, the performance improves significantly, i.e., a 4.37% improvement on the CTR dataset. Further increases from T = 1 to T = 3 continue to yield consistent gains across all three datasets. However, these gains come at the cost of reduced inference speed, as the number of recurrent steps increases. To balance accuracy and efficiency, we adopt T = 1 as the final configuration. This setting provides the best trade-off, achieving notable performance improvements while maintaining practical inference speed.

---

### Note · Authors · 2025-08-12

Dear Area Chairs and reviewers,

We sincerely appreciate the reviewers’ thoughtful feedback, which has been invaluable in enhancing the quality of our work.

**Advantages.** This work studies multi-query scene text retrieval with a box-free paradigm, which **addresses practical challenges** (w1db) of the costly boxes and unified multi-query processing. We propose Progressive Vision Embedding to capture fine-grained scene text features, which is appreciated by the reviewers as **well-motivated** (5CKX, w1db), **interesting** (EPGg, w1db), and **effective** (EPGg, 5CKX). The **extensive** evaluation (EPGg) demonstrates the **superiority** of our method (Qggy, EPGg, w1db). Moreover, the MQTR dataset enables more **realistic evaluation** (Qggy, 5CKX) and **fosters further research** (Qggy).

**Discussions.** Although the reviewers initially raised concerns, we have conducted experiments and analyses to thoroughly address them. The key points are summarized below:

1. The experiments and analyses have been added to demonstrate the effectiveness of re-ranking.
2. The evaluation protocol details have been provided for the comparisons with text spotters.
3. The computational experiments have shown the inference efficiency. Our method shows comparable performance with the advanced TG-Bridge while runs twice faster.
4. Additional experiments have been performed as requested: query style recognition module, evaluation on small-scale scene text, and quantitative analysis of mask quality.
5. A detailed analysis has been added about the challenge of extremely small text. Although this remains an open problem for box-free methods, our work takes a concrete step forward.
6. Reviewer 5CKX raised concerns regarding the significance of scene text retrieval. We highlight that **scene text retrieval is a significant problem for both OCR and retrieval community** regarding: a) Numerous works of this domain in TPAMI, ICML, and CVPR, etc., highlight its significance; b) Evidences show that both OCR and scene text retrieval remain challenging for state-of-the-art MLLMs and general retrievers; c) Scene text retrieval is a unique, object-level cross-modal retrieval task with wide applications and strong parallels to other specialized tasks (e.g., person re-ID) accepted at NeurIPS.

The details are provided below. We believe the majority of the concerns have been addressed. We thank all reviewers once again for their valuable comments and efforts.

Best regards,

Paper 20746 Authors

---

### Decision · Program_Chairs · 2025-09-17

**Decision:**

Accept (poster)

**Comment:**

This paper presents a box-free scene text retrieval method that eliminates the need for bounding box annotations by employing progressive vision embedding and style-aware queries to enhance text representation and vision-language alignment. It also introduces a new dataset to serve as a benchmark for multi-query retrieval. The authors report convincing performance gains across seven public datasets for both scene text retrieval and scene text spotting tasks. The authors provided a strong rebuttal that effectively addressed most of the concerns raised by the reviewers. The paper received three "Borderline Accept" scores and one "Borderline Reject." The reviewer who assigned the "Borderline Reject" expressed the view that scene text retrieval is a very specific and narrow task within the broader retrieval community, which may limit its interest for a general audience. After carefully reviewing all comments, the rebuttal, and the discussion, the AC recommends acceptance.